# Regulation of BRCA1 stability through the tandem UBX domains of isoleucyl-tRNA synthetase 1

Scisung Chung ®[1], Mi-Sun Kang ®[2], Dauren S. Alimbetov[3], Gil-Im Mun ®[4], Na-Oh Yunn ®[5], Yunjin Kim ®[1], Byung-Gyu Kim ®[2], Minwoo Wie[2], Eun A. Lee ®[2], Jae Sun Ra ®[2], Jung-Min Oh[6], Donghyun Lee ®[7], Keondo Lee ®[7], Jihan Kim ®[1], Seung Hyun Han[1], Kyong-Tai Kim[1], Wan Kyun Chung[7], Ki Hyun Nam ®[8,9], Jaehyun Park ®[10], ByungHoon Lee ®[11], Sunghoon Kim ®[12], Weixing Zhao[3], Sung Ho Ryu ®[1], Yun-Sil Lee ®[4], Kyungjae Myung ®[2,13] ✉ & Yunje Cho ®[1] ✉

Aminoacyl-tRNA synthetases (ARSs) have evolved to acquire various additional domains. These domains allow ARSs to communicate with other cellular proteins in order to promote non-translational functions. Vertebrate cytoplasmic isoleucyl-tRNA synthetases (IARS1s) have an uncharacterized unique domain, UNE-I. Here, we present the crystal structure of the chicken IARS1 UNE-I complexed with glutamyl-tRNA synthetase 1 (EARS1). UNE-I consists of tandem ubiquitin regulatory X (UBX) domains that interact with a distinct hairpin loop on EARS1 and protect its neighboring proteins in the multi-synthetase complex from degradation. Phosphomimetic mutation of the two serine residues in the hairpin loop releases IARS1 from the complex. IARS1 interacts with BRCA1 in the nucleus, regulates its stability by inhibiting ubi-quitylation via the UBX domains, and controls DNA repair function.

Aminoacyl-tRNA synthetases (ARSs) catalyze the ligation of each amino acid to the 3' hydroxyl group of the cognate tRNA and thereby establish the genetic code for protein synthesis[1]. In addition to their catalytic role in translation, vertebrate ARSs are involved in the regulation of various types of cell signaling, which function in cell cycle regulation, tumor suppression, cytokine activity, RNA splicing, cell adhesion, and angiogenesis[2,3]. These non-catalytic orthogonal functions are primarily achieved through various unique domains (UNEs) of ARSs acquired during evolution[4]. Spliceosome analyses identified a number of ARS variants with null catalytic activities that retain important cellular functions through UNEs, demonstrating the significance of the non-translational functions of UNEs[5]. ARSs assemble into a multi-synthetase complex (MSC), which is comprised of nine ARSs (arginyl- (RARS1), aspartyl- (DARS1), glutaminyl- (QARS1), glutamyl-prolyl- (EPRS1), isoleucyl- (IARS1), leucyl- (LARS1), lysyl- (KARS1), and methionyl-tRNA synthetase 1 (MARS1)) and three auxiliary proteins (ARS-interacting multifunctional proteins, AIMP1 (p43), AIMP2 (p38), and AIMP3 (p18)) in vertebrates[2,6,7]. Within the MSC, each ARS

[1]Department of Life Sciences, Pohang University of Science and Technology, Pohang, Republic of Korea. [2]Center for Genomic Integrity, Institute for Basic Science (IBS), Ulsan 44919, Republic of Korea. [3]Department of Biochemistry and Structural Biology, University of Texas Health San Antonio, San Antonio, TX 78229, USA. [4]Graduate School of Pharmaceutical Sciences, Ewha Womans University, Seoul, Republic of Korea. [5]Postech Biotech Center, Pohang University of Science and Technology, Pohang, Republic of Korea. [6]Department of Oral Biochemistry, School of Dentistry, Pusan National University, Pusan, Republic of Korea. [7]Department of Mechanical Engineering, Pohang University of Science and Technology, Pohang, Republic of Korea. [8]Division of Biotechnology, Korea University, Seoul, Republic of Korea. [9]Institute of Life Science and Natural Resources, Korea University, Seoul, Republic of Korea. [10]Pohang Accelerator Laboratory, Pohang University of Science and Technology, Pohang, Republic of Korea. [11]Department of New Biology, Daegu Gyeongbuk Institute of Science and Technology (DGIST), Daegu, Republic of Korea. [12]Institute for Artificial Intelligence and Biomedical Research, College of Pharmacy, Gangnam Severance Hospital, Yonsei University, Incheon 21983, Republic of Korea. [13]Department of Biomedical Engineering, Ulsan National Institute of Science and Technology, Ulsan 44919, Republic of Korea. ✉e-mail: kmyung@ibs.re.kr; yunje@postech.ac.kr

communicates with other ARSs through interactions between the UNEs and between the UNEs and catalytic (canonical or anticodon-binding) domains, and through the three auxiliary proteins. Although the exact role of the MSC remains elusive, it has been proposed to provide a channel for the efficient delivery of charged tRNAs to ribosomes[8,9]. A recent study suggests that the MSC does not affect global translation, but is critical for the nuclear localization of ARSs, and ARSs released from the MSC contribute to various cellular events[10–15]. The MSC undergoes dynamic structural changes under different conditions, and various forms of the MSC are expected to be present[16–18].

IARS1 is one of the least characterized ARSs in the MSC. It is a class 1a ARS and has an UNE-I domain at its C-terminal end that is formed by tandem ~90 residue repeats[19]. Systematic depletion and yeast two-hybrid (Y2H) analyses suggest that IARS1 binds to the WHEP domain of EPRS1 through its UNE-I domain[20,21]. However, crosslinking and mass spectrometry (XL-MS) analyses revealed that IARS1 interacts with RARS1, LARS1, and MARS1 in the MSC[18]. These results suggest the dynamic and multi-valent interactions of IARS1 with the components of MSC. Furthermore, little is known about the orthogonal function of IARS1 UNE-I. Upregulation of the gene encoding IARS1 increases p38 MAPK pathway activation and reduces PI3K pathway activation, leading to phenotypic switching and apoptosis in aorta vascular smooth muscle cells[22]. However, it is unknown whether these functions are correlated with the UNE-I domain.

In this study, we determined the structure of the chicken EARS1–IARS1 complex. We show that UNE-I is formed by tandem repeats of ubiquitin regulatory X (UBX) domains and that IARS1 interacts with a distinct hairpin loop in the EARS1 anticodon-binding domain via its tandem UBX domains. We also show that phospho-mimetic mutation of one or two serine residues in the EARS1 hairpin loop releases IARS1 from the complex. IARS1 UBX domains interact with the heterodimeric RING domains of BRCA1-BARD1 and the BRCT domain of BRCA1 and protect the complex from ubiquitin (Ub)-mediated degradation. We show that the IARS1 UNE-I domain participates in DNA repair via maintaining the stability of the BRCA1-BARD1 complex.

## Results

### Interaction between EPRS1 and IARS1

Systematic depletion analysis, Y2H, and XL-MS provided different results regarding IARS1-interacting proteins in the MSC[18,20,21]; therefore, we performed mass spectrometry analysis using cells expressing wild-type IARS1, C-terminal-deleted IARS1 (ΔC, residues 1–1081), and the IARS1 UNE-I domain (residues 942–1262) alone. While wild-type IARS1 most extensively captured EPRS1, it also interacted with LARS1, QARS1, RARS1, and DARS1. However, IARS1ΔC failed to interact with other ARSs except LARS1 (Fig. 1a). By contrast, the UNE-I domain alone associated with EPRS1, QARS1, RARS1, and DARS1, but did not interact with LARS1. This suggests that IARS1 interacts with EPRS1 and LARS1 through the UNE-I and catalytic domains, respectively. IARS1 bound to the UNE-L domain (residues 1062–1176) of LARS1 (Supplementary Fig. 1a). EPRS1 is a chimera of EARS1 and PARS1 connected by three copies of the WHEP domain[12]. Thus, we further analyzed which parts of EPRS1 and IARS1 are responsible for their binding. Purified EARS1 with three WHEP motifs (residues 1–1015) directly associated with full-length IARS1, and removal of the three WHEP motifs did not affect complex formation (Fig. 1b). By contrast, removal of the second repeat in UNE-I (residues 1151–1262) of IARS1 abrogated the interaction with EARS1 (Fig. 1b). Collectively, pull-down, size exclusion chromatography, and immunoprecipitation analyses revealed that EARS1, IARS1, and LARS1 form a ternary complex in which IARS1 binds to the catalytic domain of EARS1 via the second repeat in UNE-I and LARS1 binds to the catalytic domain of IARS1 through the C-terminal UNE-L (Fig. 1a–c and Supplementary Fig. 1a–c).

### Overall structure of the EARS1–IARS1 complex

To elucidate the molecular basis of the interplay between EARS1 and IARS1, we attempted to determine the EARS1–IARS1 structure. Most of our efforts to determine the full-length EARS1–IARS1 complex were unsuccessful; therefore, we employed two strategies to aid the crystallization. First, we used the EARS1–IARS1 complexes from five different species. Second, we fused the UNE-I of IARS1 to the C-terminal end of the EARS1 construct that can be cleaved by a specific protease after initial purification. This is because there was a contaminant protein that could not be removed when EARS1 and IARS1 UNE-I were co-expressed. The purified EARS1–IARS1 UNE-I fusion was subjected to further proteolytic digestion by subtilisin, which generated two fragments, EARS1 and IARS1 UNE-I (Fig. 1c). We obtained showers of thin needle crystals of *Gallus gallus* EARS1 (GgEARS1; residues 176–706) complexed with the IARS1 C-terminal domain (residues 965–1265) containing UNE-I, which diffracted at no better than 3.5–4 Å resolution. While we could not improve the size and quality of the crystals, we determined the structure of the GgEARS1–IARS1 complex at 2.4 Å resolution using the Pohang Advanced Light Source X-ray free electron laser (PAL-XFEL, Supplementary Table 1). We also determined the apo GgEARS1 (residues 176–706) structure at 2.5 Å resolution (Supplementary Note and Supplementary Table 1). The overall structure of GgEARS1 consists of an N-terminal catalytic domain (acceptor end-binding and dinucleotide-binding domains) and C-terminal anticodon-binding domain (proximal β-barrel (Pβ) and distal β-barrel (Dβ) domains) (Fig. 1d and Supplementary Fig. 2, 3a). The helical subdomain (stem-binding domain) connects the catalytic and anticodon-binding domains. The overall structure of GgEARS1 most resembles bacterial (PDB 1QTQ) or human QARS1[23,24] (PDB 4R3Z; Supplementary Fig. 3a–d). The electron density map of both the Pβ and Dβ domains showed significant disorder in the apo GgEARS1 structure, indicating that the regions are highly mobile (Supplementary Fig. 2a). However, these regions become ordered upon binding to the IARS1 C-terminal domain, which consists of three repeats of a β-grasp domain (Fig. 1d–f and Supplementary Fig. 2b). GgEARS1 contains a distinct hairpin loop unique to vertebrate EARS1s in the Pβ domain (Figs. 1d and 2a–c). This loop is located at the opposite face of the tRNA-binding surface of EARS1 and is oriented perpendicular to the longest axis of GgEARS1 (Supplementary Fig. 3a, d). UNE-I binds to the hairpin loop and nearby surface of EARS1 via the second and third β-grasp domains.

### Structural differences between vertebrate EARS1 and QARS1

Because the GgEARS1 structure is the first eukaryotic EARS1 structure, we briefly compared it with those of other related ARSs. Overall, each domain of GgEARS1 is very similar to the corresponding domain of QARS1 or archaeal EARS (PDB 3AII), except the Dβ domain, in which significant differences are observed between GgEARS1 and archaeal EARS[23–25] (Supplementary Fig. 3a–c). The largest differences between GgEARS1 and bacterial or human QARS1 are observed in the following four loops (Supplementary Fig. 3d–g): (i) a distinct hairpin loop in the Pβ domain in GgEARS1; the corresponding loop is much shorter and wider in QARS1, which prevents binding of IARS1 UBX2 (Supplementary Fig. 3a, c); (ii) the S17–S18 loop proximal to U33 and C34 of the anticodon-binding site of a modelled tRNA in GgEARS1 (Supplementary Fig. 3e); (iii) the absence of the loop near the G36 base in GgEARS1 (Supplementary Fig. 3f); and (iv) the absence of the four-residue β-turn that wedges the acceptor-binding loop in GgEARS1 (Supplementary Fig. 3g). The latter three features may contribute to the selectivity for binding of tRNA[Glu] over tRNA[Gln] by vertebrate EARS1. These differences are more extensively described in Supplementary Note.

### The IARS1 UNE-I folds into tandem UBX domains

Each β-grasp repeat in the IARS1 C-terminal domain consists of one or two helices packed against four or five stranded antiparallel β-sheets. The first β-grasp domain corresponds to the KH domain, whereas the

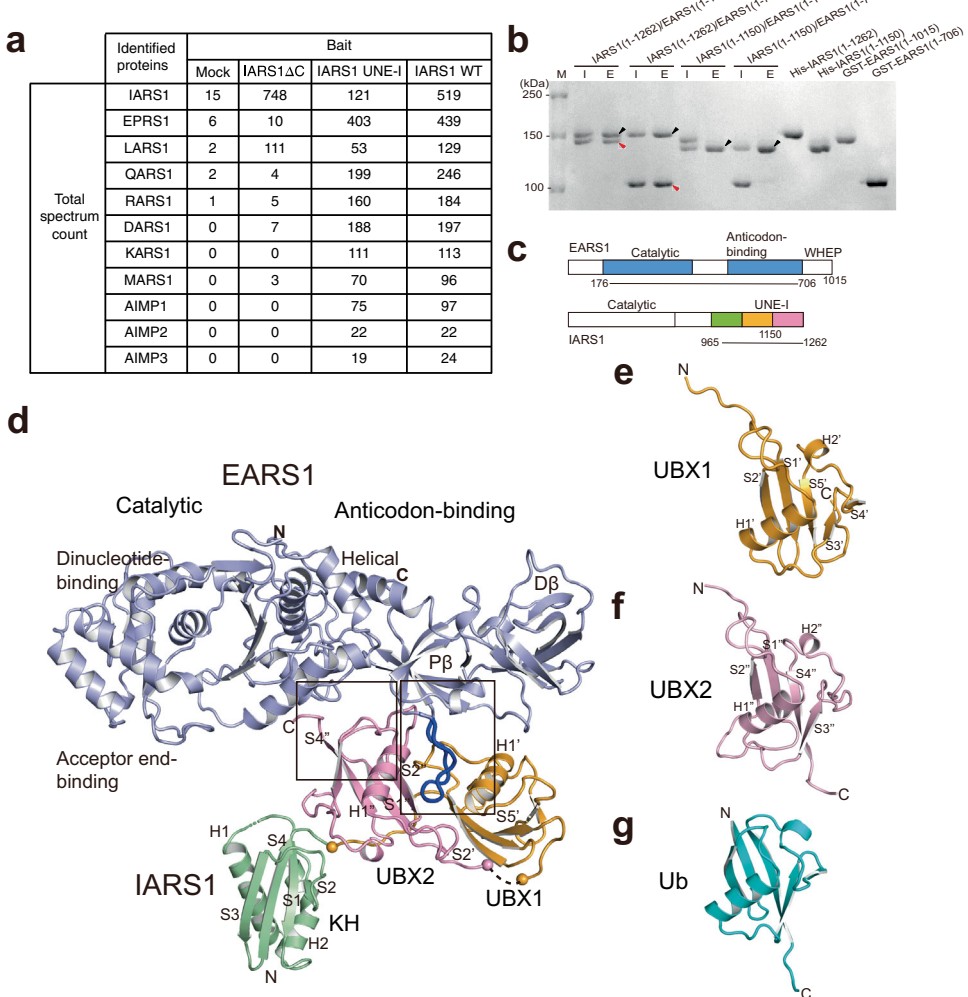

**Fig. 1 | Overall structure of the EARS1–IARS1 complex. a** A list of MSC components identified by mass spectrometry. Total spectrum count of identified proteins interacting with IARS1ΔC, IARS1 UNE-I, or IARS1 WT is shown. **b** Affinity pull-down analysis of the interaction between EARS1 and IARS1. The input (I) and eluate (E) fractions were analyzed by SDS-PAGE and Coomassie Blue staining. Black and red arrowheads indicate eluted His-IARS1 and GST-EARS1, respectively. The data are representative of three independent experiments. Source data are provided as a Source Data file. **c** A schematic illustration of the architectures of EARS1 and IARS1. **d** Structure of the EARS1–IARS1 complex. The core regions of EARS1 (light blue) and IARS1 UNE-I (KH, green; UBX1, orange; and UBX2, pink) are shown at the top and bottom of the image, respectively. Close-up views of the boxed regions are shown in Fig. 2a–c. **e–g** Structures of UBX1 **e**, UBX2 **f**, and Ub **g**.

second and third β-grasp domains most resemble Ub or the UBX domain with a rms deviation value of 1.8 and 2.0 Å, respectively, for 76 Cα residues, which we termed UBX1 and UBX2[26–28] (Fig. 1e–g). The second and third β-grasp domains lack two glycine residues at the end, a hallmark of Ub. The two repeats share high structural similarity with a rms deviation value of 1.4 Å. The UBX1 and UBX2 domains are connected by an extended 10-residue loop with a length of 30 Å, and the KH and UBX1 domains are connected by an extended 15-residue loop (30 Å). Together, the extended C-terminal domain (residues 965–1262) contains one KH domain and tandem repeats of the UBX domain with dimensions of 30 × 52 × 25 Å (Fig. 1d and Supplementary Fig. 4a–c).

In the UBX domain, we defined the face of the five stranded sheets as the "exposed face" and the helix face as the "obscured face" (Supplementary Fig. 4a). The two UBX repeats are arranged in an antiparallel orientation, in which the second strand of the sheet in each UBX domain faces the opposite direction to the other strands. The 2-fold axis is positioned between the S2′ and S2″ strands perpendicular to the strands (Supplementary Fig. 4a). UBX1 and UBX2 interact with each other through the S2′ and S2″ strands, between the S2′–H1′ loop and S2″ strand, and between the S2′ strand and S2″–H1″ loop. Upon packing, a wide groove with a width of 15 Å, height of 10 Å, and depth

of 11 Å is formed between the first helices of UBX1 and UBX2 (Supplementary Fig. 4a, c). One of the proposed functions of UBX is to mediate interactions between different protein partners. We searched for conserved surface residues in IARS1 UBX domains and identified a set of conserved residues that interact with EARS1 in the wide groove between UBX1 and UBX2 (Supplementary Fig. 4b, c).

### IARS1 binds to the EARS1 hairpin loop through its tandem UBX domains

The UBX1 and UBX2 domains both directly interact with EARS1 via the groove between the inter-helices at the obscured face (Fig. 1d and Supplementary Fig. 4a). A total of 2429 Å² of surface area becomes buried upon complex formation. Complex formation between EARS1 and IARS1 is largely achieved through the two interfaces. The first interface is formed between UBX2 and the helical subdomain (stem-binding domain) of EARS1 (Figs. 1d and 2a). In this interface, the hydrophilic interaction between the C-terminal loop of UBX2 and EARS1 is extensive, with seven residues of UBX2 interacting with eight residues of EARS1 (Fig. 2a). The second interface is defined between the two UBX domains and the extended hairpin loop (22 Å long, residues Tyr684 to Pro696) of the EARS1 Pβ domain (Fig. 2b, c and

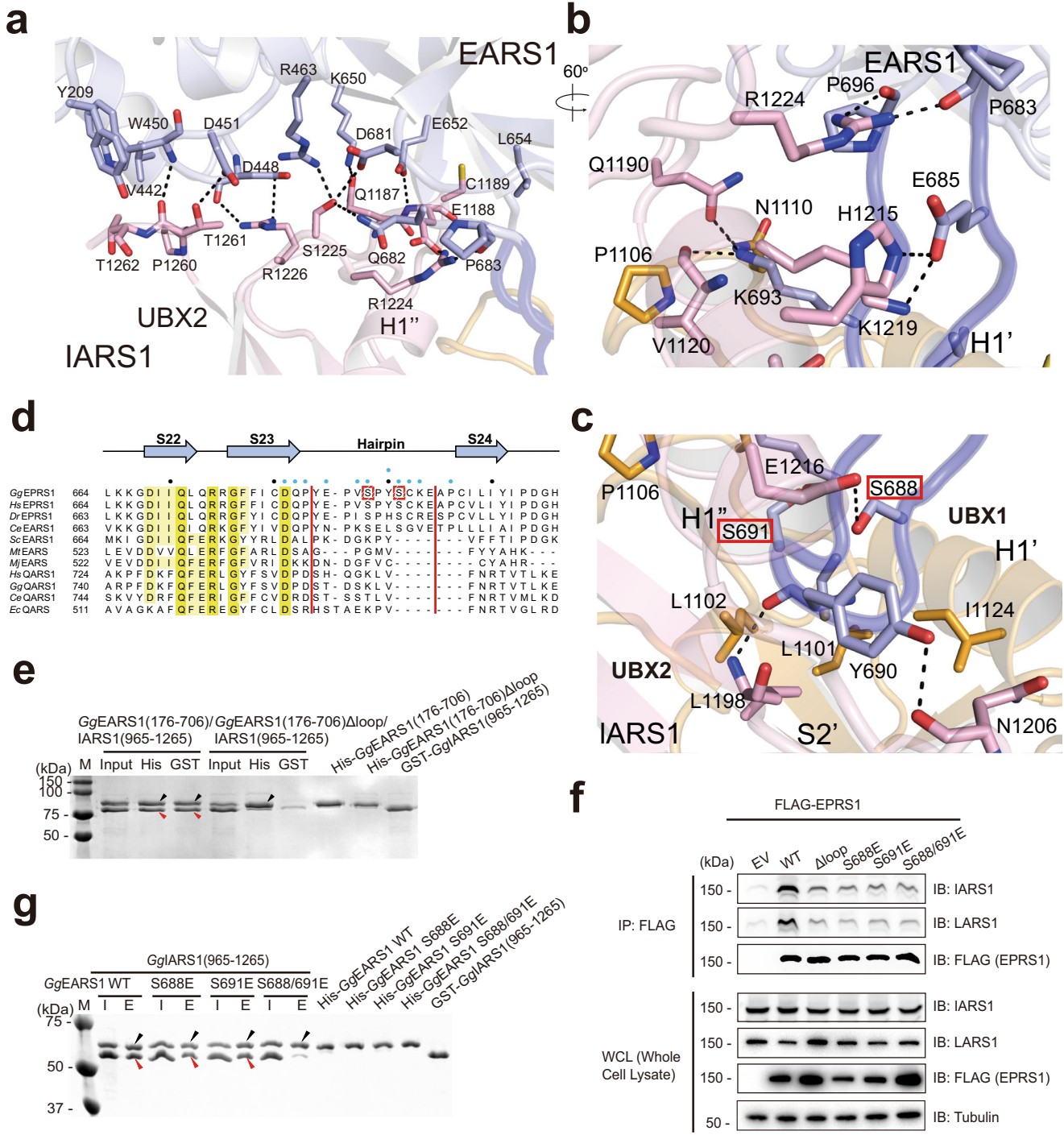

**Fig. 2 | Interaction between the EARS1 hairpin loop and IARS1 UBX domains.**
**a** Close-up view of the interface between EARS1 and IARS1 UBX2. **b, c** Close-up view of the interaction between the EARS1 hairpin loop and IARS1 UBX domains. Mutated residues in the pull-down analysis are indicated by red boxes. **a**–**c** Color schemes are same as those in Fig. 1d. H-bonds and ion-pairs are shown in black dots. **d** Structure-based sequence alignments of EARS1 and QARS1 orthologs. The secondary structure of *Gg*EARS1 is displayed on top of the sequences. IARS1-contacting residues are indicated by blue circles. Every tenth residue is marked with a black dot. **e** In vitro pull-down assay of *Gg*EARS1 and the IARS1 UNE-I domain. His-*Gg*EARS1 or His-*Gg*EARS1Δhairpin was pulled down with Ni-NTA resin, and co-precipitated GST-*Gg*IARS1 was detected by Coomassie staining. Black and red

arrowheads indicate eluted His-*Gg*EARS1 and GST-*Gg*IARS1, respectively. **f** Incorporation of EPRS1 phosphomimetic mutants into the MSC. Flag-tagged, full-length EPRS1 wild-type protein and the phosphomimetic (Ser-to-Glu, S-E) mutant were expressed in HEK293T cells by transient transfection. Lysates were immuno-precipitated with an anti-Flag antibody, and co-precipitated IARS1 or LARS1 was detected by immunoblotting with an anti-IARS1 or anti-LARS1 antibody. **g** Wild-type or phosphomimetic mutant His-*Gg*EARS1 was pulled down with Ni-NTA resin, and co-precipitated GST-*Gg*IARS1 was detected by Coomassie staining. Black and red arrowheads indicate eluted His-*Gg*EARS1 and GST-*Gg*IARS1, respectively. **e**–**g** The data are representative of three independent experiments. Source data are provided as a Source Data file.

Supplementary Fig. 5a). One face of the EARS1 hairpin forms extensive H-bonds with residues of UBX2. On another face of the hairpin loop, hydrophobic interactions are dominant with UBX1, but additional interactions are also observed with the S2'–H1' loop of UBX1 (Fig. 2b, c and Supplementary Fig. 5b).

To verify the importance of the EARS1 hairpin loop in the EARS1-IARS1 complex, we removed 11 residues (residues 684–694) from the hairpin loop and examined binding between EARS1 and IARS1 (Fig. 2d). Hairpin-deleted EARS1 (EARS1Δloop) failed to interact with IARS1 (Fig. 2e and Supplementary Fig. 6). We also expressed the Flag-tagged EPRS1 mutant lacking the hairpin loop in human embryonic kidney 293T (HEK293T) cells. Co-immunoprecipitation (co-IP) analysis showed that the EPRS1Δloop mutant weakly interacted with IARS1 (Fig. 2f, lane 3). EPRS1 forms a dimer via PARS1 and therefore a portion of the EPRS1 mutant could form a dimer with endogenous EPRS1, which may explain this residual interaction.

## Phosphomimetic mutation of Ser in the hairpin loop dissociates IARS1 from EARS1

At the tip of the hairpin loop, two conserved Ser (Ser688 and Ser691) and one Tyr (Tyr690) residues are present. Although Ser691 is not involved in the direct interaction with the UBX domains, Ser688 makes contact with Glu1216 of UBX2, whereas Tyr690 interacts with Leu1198 and Asn1206 (Fig. 2c). In the PhosphoSitePlus database, Ser688, Tyr690, and Ser691 are assigned as putative phosphorylation sites[29]. To investigate whether phosphorylation of these residues leads to release of IARS1 from EARS1, we generated single or double phosphomimetic Ser-to-Glu mutants. While mutation of Ser688 or Ser691 to glutamate retained the interaction of EARS1 with IARS1, simultaneous mutation of both residues dissociated IARS1 from EARS1 (Fig. 2g). To further investigate the effect of phosphorylation of these residues on IARS1 release, we transfected cells with single or double phosphomimetic mutants of Flag-tagged EPRS1 and immunoprecipitated wild-type EPRS1 or the mutants. In contrast with the in vitro analysis, single mutation of EPRS1, S688E, or S691E dissociated IARS1 from the complex in vivo (Fig. 2f).

## IARS1 interacts with BRCA1 via its UBX domains

The UBX domain shares high structural similarity with Ub. Although Ub has various cellular functions, one of its important functions is to mediate protein–protein interactions and protein degradation[30]. Therefore, we hypothesized that IARS1 interacts with various cellular proteins through its UBX domains and protects these proteins from E3 ligase-mediated degradation (Fig. 3a). We first searched for binding partners of IARS1 in the Biological General Repository for Interaction Datasets (BioGRID) database[31]. A total of 158 putative binding proteins were classified by their functions, including UBL conjugation, cell cycle regulation, transcription, translation, chaperone, development, and metabolism (Fig. 3b). We next examined if depletion of IARS1 affected the stabilities of various binding proteins. We selected EPRS1 and LARS1 as known partners and USP14, BRCA1, CDK4, CDK9, and VDAC1 as putative partners, and examined the effects of IARS1 depletion on the stabilities of these proteins (Supplementary Fig. 7a, b). Knockdown of IARS1 with siRNA^IARS1#3 or 3'UTR-specific siRNA^IARS1 exhibited most pronounced effect on the stabilities of LARS1 and BRCA1. Also, siRNA^IARS1#3-mediated knock-down of IARS1 noticeably decreased the protein levels of USP14 and CDK4, and slightly decreased protein levels of EPRS1 and CDK9 (Supplementary Fig. 7a). We concluded that IARS1 interacts with LARS1 and BRCA1 and contributes to their stabilities.

To further investigate the role of IARS1 in stabilizing proteins through direct interactions, we selected BRCA1 as a target protein because stable isotope labeling mass spectrometry analyses indicate that IARS1 interacts with BRCA1[32]. To confirm the interaction between BRCA1 and IARS1, we performed co-IP analysis with endogenous BRCA1 in HCT116 cells, and demonstrated that BRCA1 interacted with

IARS1 (Fig. 3c). Although IARS1 co-immunoprecipitated with BRCA1 in cell extracts, it is unclear whether it directly associates with BRCA1. Affinity pull-down analysis revealed that purified IARS1 directly interacted with the BRCA1–BARD1 complex in vitro (Fig. 3d). To examine which region of IARS1 is responsible for the interaction with BRCA1, we performed co-IP analysis using full-length IARS1, UNE-I truncated IARS1 (IARS1ΔC), or IARS1 UNE-I (residues 942–1262) alone. A SV40 nuclear localization signal (NLS) sequence (PKKKRKV) was fused to the N-termini of these proteins to enhance their nuclear localization. While full-length IARS1 and the UNE-I domain alone co-precipitated with BRCA1, the IARS1ΔC fragment did not interact with BRCA1, suggesting that the UBX domains are responsible for binding of IARS1 to BRCA1 (Fig. 3e).

Both the N-terminal RING and C-terminal BRCT domains of BRCA1 can mediate protein–protein interactions[33–37]; therefore, we analyzed which part of BRCA1 is responsible for the binding of IARS1. Because the RING domain of BRCA1 alone is unstable in an isolated state, we used the BRCA1-BARD1 complex for the expression and purification of the heterodimeric RING domains. In vitro pull-down analysis showed that while the heterodimeric RING (residues 1–304)-RING (residues 26–142) domains of the BRCA1-BARD1 complex and the BRCT domain (residues 1645–1863) of BRCA1 interacted with IARS1, the BRCT domain (residues 566–777) of BARD1 did not (Fig. 3f). Consistent with this observation, the BRCA1 complex lacking either RING or BRCT domain interacted with IARS1 (Fig. 3g).

## Nuclear localization of IARS1

To interact with BRCA1, IARS1 must localize to the nucleus. Previous analyses suggest that a small amount (approximately 2%) of active IARS1 localizes to the nucleus and that IARS1 is primarily located in the cytoplasm[38]. To further investigate the nuclear localization of IARS1, the distribution of IARS1 was analyzed by immunofluorescence microscopy and cell fractionation (Supplementary Fig. 8a, b). Overexpressed IARS1 was clearly detected in the nucleus by confocal microscopy using an anti-Flag antibody. Cell fractionation to separate nuclear and cytoplasmic components revealed that about 7% of IARS1 localized to the nucleus upon ectopic overexpression. These observations were made under normal conditions, and further analysis is required to determine whether the nuclear fraction of IARS1 increases in response to a specific cue.

We next examined the interaction between endogenous IARS1 and BRCA1 in the nucleus. We first isolated the nuclear fractions of HeLa cells. Co-IP analysis of BRCA1 using a BRCA1-specific antibody in the nuclear extract revealed that BRCA1 captured endogenous IARS1, which further supports the interaction between BRCA1 and IARS1 (Fig. 3h).

## IARS1 depletion leads to BRCA1 degradation

Next, to examine the effects of IARS1 binding to BRCA1, we depleted IARS1 in HeLa cells using four siRNAs that targeted distinct regions of the IARS1 sequence. Three siRNAs targeted the coding region of the mRNA (siRNA^IARS1#1 to #3) and one targeted the 3' untranslated region (3'UTR). All siRNAs significantly reduced the levels of BRCA1 protein and the IARS1-binding protein LARS1 (Fig. 4a and Supplementary Fig. 7a, b). Knock-down of IARS1 using siRNA^IARS1#1 and #2 exhibited more notable effect in reducing the levels of BRCA1 than siRNA^IARS1#3 and 3'UTR-specific siRNA^IARS1. To rule out the possibility that BRCA1 is transcriptionally downregulated by siRNA^IARS1, we performed real-time polymerase chain reaction (PCR) analysis of BRCA1. siRNA^IARS1#3 and 3'UTR-specific siRNA^IARS1 did not affect the mRNA level of BRCA1 (Supplementary Fig. 7c, d). siRNA^IARS1#1 and #2 decreased the mRNA expression level of BRCA1 to 30–40% with respect to scrambled siRNA and decreased the protein level of BRCA1 significantly more than scrambled siRNA (Fig. 4a and Supplementary Fig. 7e). To confirm that the effect of IARS1 depletion on the BRCA1 level was not due to

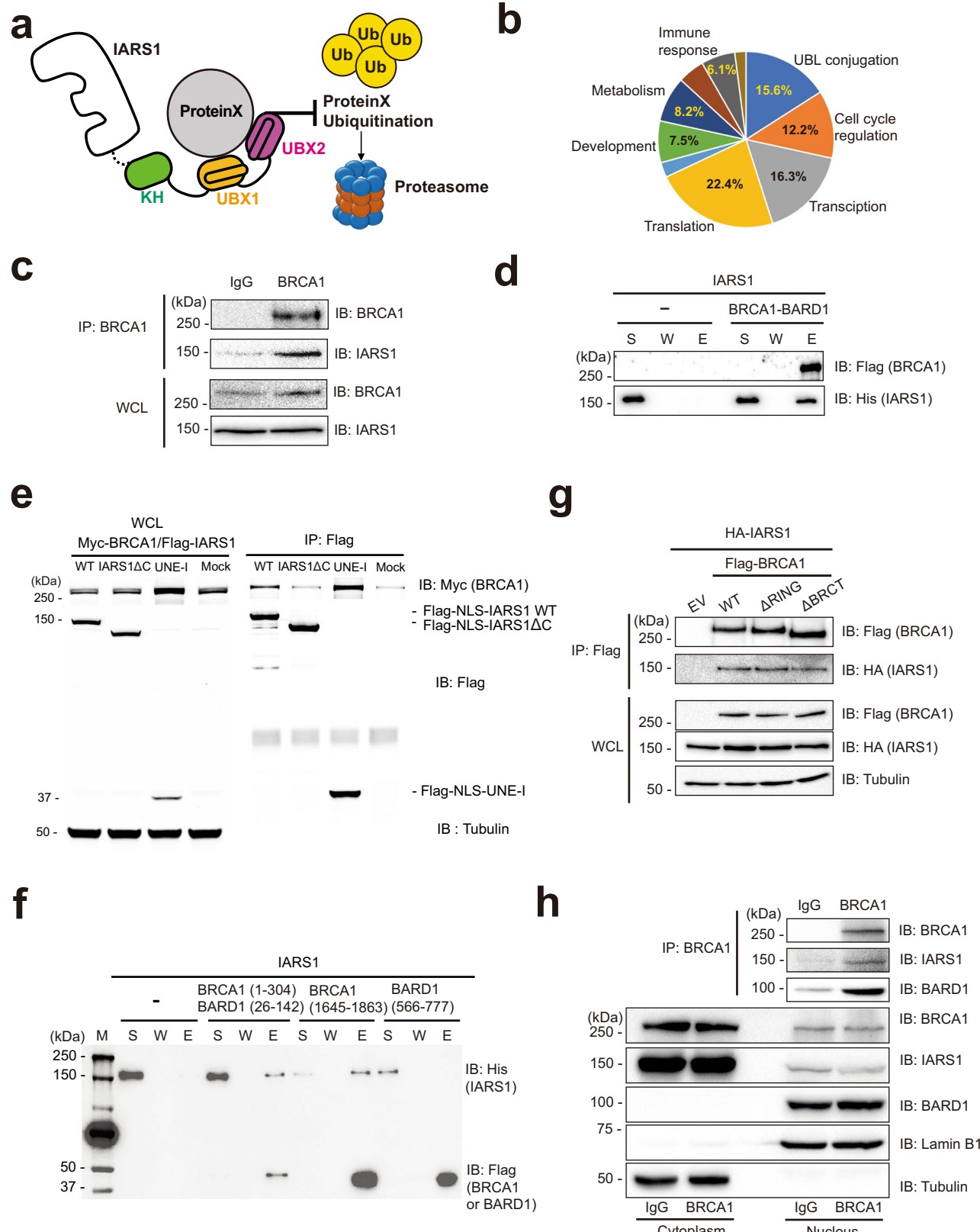

inhibition of de novo protein synthesis, we immunoblotted IARS1-depleted cell lysates with an anti-puromycin antibody. Puromycin incorporation during protein synthesis reflects the rate of mRNA translation in vitro; therefore, an antibody against puromycin can be used to monitor global protein synthesis[39]. The total amount of puromycin remained constant, indicating that knockdown of IARS1 does

not affect global translation (Supplementary Fig. 9a, b). These results suggest that IARS1 post-translationally regulates the protein level of BRCA1.

We then investigated the role of the IARS1 UBX domains in stabilizing BRCA1. We reintroduced full-length NLS-IARS1, -IARS1ΔC, or -IARS1 UNE-I fragment genes into cells treated with 3′UTR-specific

**Fig. 3 | The UBX domains of IARS1 interact with the RING and BRCT domains of BRCA1 for protein stability. a** Schematic illustration of the hypothetical model by which IARS1 UBX domains protect partner proteins from proteasomal degradation. **b** The candidate proteins that interact with IARS1 were classified into several biological processes using the BioGrid database. **c** Immunoprecipitation of endogenous BRCA1 was performed with an anti-BRCA1 antibody in HCT116 cells. Co-precipitated endogenous IARS1 was detected by immunoblotting with an anti-IARS1 antibody. **d** In vitro affinity pull-down analysis of IARS1 with the BRCA1-BARD1 complex via the Flag tag of BRCA1. The supernatant (S), wash (W), and eluate (E) fractions were analyzed by immunoblotting with anti-Flag and anti-His antibodies. **e** Immunoassay of co-expressed Myc-BRCA1 and Flag-tagged full-length IARS1 or its various domains in HEK293T cells. Flag-IARS1 was immunoprecipitated with an anti-

Flag antibody, and co-precipitated Myc-BRCA1 was detected by immunoblotting with an anti-Myc antibody. **f** In vitro pull-down assay of IARS1 and the following fragments of the BRCA1-BARD1 complex: RING-RING domains, BRCT domain of BRCA1, and BRCT domain of BARD1. **g** HEK293T cells were co-transfected with wild-type Flag-tagged BRCA1 or deletion mutants (ΔRING or ΔBRCT domain) and HA-tagged IARS1, and then immunoprecipitation was performed with an anti-Flag antibody. Co-precipitated IARS1 was detected by immunoblotting with an anti-HA antibody. **h** Co-IP analysis for the interaction between endogenous IARS1 and BRCA1 using a BRCA1-specific antibody in nuclear extract. **c–h** The data are representative of three independent experiments. Source data are provided as a Source Data file.

siRNA[IARS1]. Wild-type NLS-IARS1 restored the reduced protein levels of BRCA1 (Fig. 4b). Introduction of the NLS-IARS1 UNE-I domain alone also restored the protein levels of BRCA1 in a dose-dependent manner. To further determine whether IARS1 regulates BRCA1 degradation, IARS1-depleted HeLa cells were treated with cycloheximide and the stability of BRCA1 was investigated. Depletion of IARS1 decreased the stability of BRCA1 in comparison with control cells (Fig. 4c). BRCA1 stability is reportedly regulated by Ub-dependent proteolysis[40,41]. We hypothesized that IARS1 prevents ubiquitylation of BRCA1 through a competitive interaction of its UBX domains and examined the effect of IARS1 on ubiquitylation of BRCA1. BRCA1 was ubiquitylated upon depletion of IARS1, and its stability was decreased. By contrast, in the presence of IARS1, significantly less BRCA1 was ubiquitylated and the level of BRCA1 was normal, which suggests that IARS1 protects BRCA1 from Ub-mediated degradation (Fig. 4d, Supplementary Fig. 10a–d). The reduced BRCA1 level upon IARS1 depletion was restored by treatment with the proteasome inhibitor MG132 (Fig. 4d, Supplementary Fig. 10e, f). We also examined the stability of BRCA1 in the presence of the UBX domain alone. Overexpression of full-length IARS1 decreased ubiquitylation of BRCA1 and increased its stability (Fig. 4e, Supplementary Fig. 10b, d). However, ubiquitylation and stability of BRCA1 upon overexpression of IARS1 lacking the UNE-I domain were similar to those in the control. Expression of IARS1 UNE-I alone decreased ubiquitylation and increased the stability of BRCA1 to similar levels as full-length IARS1 (Fig. 4e, Supplementary Fig. 10b, d). These results suggest that BRCA1 degradation is primarily regulated by the UNE-I domain of IARS1 and that this domain is essential for maintenance of BRCA1 stability.

## IARS1 depletion leads to impaired DNA repair

Because IARS1 controls the stability of BRCA1, we investigated whether IARS1 depletion affects the DNA repair pathway, in which BRCA1 plays a critical role[34]. We measured the efficiencies of homologous recombination (HR) and non-homologous end joining (NHEJ) in cells treated with siRNA[IARS1]#1 and #2. We have used siRNA[IARS1]#1 and #2 because they most significantly reduced the level of BRCA1 protein among the four siRNAs we examined (Fig. 4a, Supplementary Fig 7a, b). Direct-repeat green fluorescent protein (DR-GFP) and EJ5-GFP reporters were used to measure double-strand break (DSB)-induced HR and multiple classes of NHEJ events, respectively[42,43]. Knockdown of endogenous IARS1 using siRNA reduced the HR efficiency by approximately 60% but did not affect NHEJ, consistent with the role of BRCA1 in HR (Fig. 5a, b, Supplementary Fig 11a, b). Next, we investigated whether down-regulation of BRCA1 upon IARS1 depletion affects DNA damage-induced assembly of RAD51 nuclear foci, an important early event in HR[44]. Notably, knockdown of endogenous IARS1 reduced RAD51 focus formation by approximately 50–70% upon ionizing radiation (IR) exposure, consistent with the previous finding that reduction of BRCA1 impairs RAD51 recruitment to DSBs[44] (Fig. 5c). Increased formation of γH2AX foci upon exposure to IR in cells with or without IARS1 depletion indicated that IARS1 depletion did not affect induction of DSBs (Fig. 5d).

We next quantified single-stranded DNA (ssDNA) at DSB sites using the ER-*AsiSI* system under IARS1 depletion conditions to measure resection adjacent to specific DSBs[45]. DSB resection involves an initial short-range resection catalyzed by the MRE11-RAD50-NBS1-CtIP complex generating 3′-ssDNA near the DSB end, which is further elongated by EXO1 and DNA2[46]. Depletion of IARS1 significantly reduced resection of DSB ends comparable with depletion of CtIP (Fig. 5e). Inactivation of BRCA1 triggers a moderate decrease in ssDNA levels compared with that induced by abrogation of CtIP[47–49]. Therefore, we investigated whether IARS1 depletion affects the protein levels of CtIP or essential resection factors such as EXO1. Notably, IARS1 depletion decreased the protein levels of CtIP and EXO1 (Supplementary Fig. 11c). Next, we measured the amount of EXO1 bound to chromatin after exposure to IR in order to investigate its recruitment of DNA damage sites. In the presence of DNA damage, IARS1 depletion did not affect the amount of chromatin-bound EXO1 (Supplementary Fig. 11d). These results suggest that IARS1 not only contributes to BRCA1 stability but also regulates the levels of CtIP and possibly other factors involved in end resection.

We further investigated DNA damage induced by depletion of IARS1. To investigate whether IARS1 affects gross chromosomal rearrangement, metaphase spreads of IARS1-depleted cells were examined for chromosomal aberrations by Giemsa staining[50]. Chromosomes from IARS1-depleted cells had a higher incidence of end-to-end fusions, inversions, and aneuploidy than chromosomes from scrambled siRNA-treated cells; IARS1 depletion increased the frequency of sister chromatid exchange (SCE) by approximately 10% compared with control cells (Fig. 5f). Inhibition of poly-ADP ribose polymerase (PARP) activity in BRCA1-deficient cells results in synthetic lethality[51,52]. Consistently, in clonogenic cell survival assays, IARS1-depleted cells were significantly more sensitive to the PARP inhibitor olaparib than control cells (Fig. 5g). Collectively, our results suggest that IARS1 UNE-I plays an important role in DNA damage repair by stabilizing BRCA1 and additional factors that control end resection and SCE.

## Discussion

Vertebrate ARSs have evolved to have nontranslational functions, which are associated with UNE domains added to the catalytic domains[4–6]. In this study, we investigated the orthogonal function of IARS1, one of the least characterized ARSs. We revealed that IARS1 possesses tandem UBX domains at its C-terminal end and that one of its functions is to interact with cellular proteins and protect them from degradation. Among all additional domains of ARSs, this is the first example of UBX domains fused to a catalytic domain[4].

Complex formation between EARS1 and IARS1 is only observed in vertebrate MSCs[21]. Tandem UBX domains form a central groove that is optimized to recognize the extended hairpin loop. The EARS1–IARS1 interface is further augmented through the binding of UBX2 to the flat surface of the Pβ domain. Because depletion of IARS1 destabilizes EARS1 and neighboring proteins in the MSC, it is likely that the association of IARS1 with EARS1 through the UBX domains

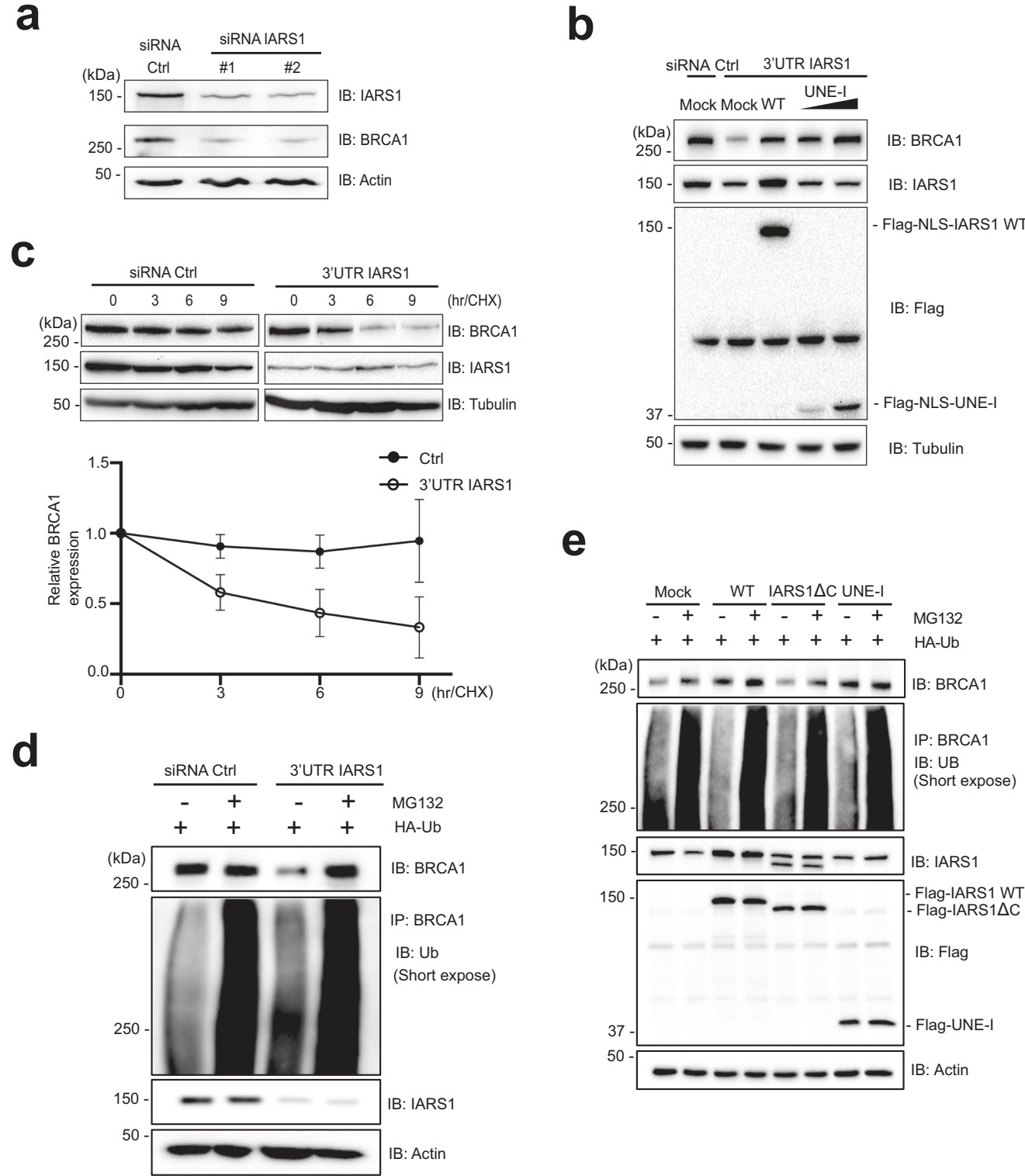

protects the MSC from cellular degradation[20]. Protection of the MSC via the UBX domains may prevent abortive functions in cellular metabolism.

The EARS1–IARS1 complex structure is supported by systematic depletion analysis, which showed that IARS1 mediates the interaction between EPRS1 and LARS1[20] (Fig. 1a and Supplementary Fig. 1a, b). Cells lacking IARS1 exhibited reduced levels of EARS1, neighboring ARSs, and AIMP2. However, our results differ from the previous findings based on XL-MS analysis that IARS1 binds to MARS1, LARS1, and RARS1 through its catalytic domain[18]. One possible explanation is that the

MSC is an intrinsically unstable complex that undergoes a dynamic structural transition under different cellular conditions, and various forms of the MSC may be present in the cell[16,20]. However, we cannot exclude the possibility that in XL-MS analysis, MSC components were pulled down by an antibody that recognizes the EPRS1 WHEP domain, which is immediately adjacent to the EARS1 hairpin loop and the antibody binding may obscure the IARS1-binding site in the hairpin loop.

Although the cellular function of the UBX domain is not clearly understood, its structural similarity with Ub suggests that it

**Fig. 4 | The UBX domains of IARS1 are responsible for BRCA1 stability. a** Protein extracts from HeLa cells transfected with control siRNA or two siRNA[IARS1] were immunoblotted with anti-IARS1, anti-BRCA1, and anti-Actin (loading control) antibodies. **b** SV40 NLS fused wild-type IARS1 or -IARS1 UNE-I (residues 942-1262) was ectopically expressed following depletion of IARS1 using 3'UTR-specific siRNA[IARS1]. **a, b** The data were representative of three independent experiments. Source data are provided as a Source Data file. **c** The stability of BRCA1 in cells with or without IARS1 depletion was assessed by cycloheximide-chase analysis. HEK293T cells were transfected with control siRNA or 3'UTR-specific siRNA[IARS1] and treated with 100 μg/mL cycloheximide. Lysates prepared at the indicated time points were immunoblotted with an anti-BRCA1 antibody. The graph shows relative levels of BRCA1 expression. Error bars indicate standard deviation (SD) of the mean ($n = 3$, independent cell cultures). **d** In vivo ubiquitylation assay of BRCA1 in transfected HEK293T cells with or without IARS1 depletion. The blot with short exposure is shown. HEK293T cells co-transfected with control siRNA or 3'UTR-specific siR-NA[IARS1] and Ub were treated with 10 μM MG132 for 4 hr and analyzed by immunoblotting with the indicated antibodies. BRCA1 immunoprecipitated with an anti-BRCA1 antibody was analyzed by immunoblotting with an anti-Ub antibody. Quantification of ubiquitylated BRCA1 and the blot with long exposure are shown in Supplementary Fig. 10a and c. **e** In vivo ubiquitylation assay of BRCA1 in HEK293T cells transfected with full-length IARS1, IARS1ΔC, and IARS1 UNE-I alone. The blot with short exposure is shown. HEK293T cells co-transfected with indicated cDNAs and Ub were treated with 10 μM MG132 for 4 h and analyzed by immunoblotting with the indicated antibodies. BRCA1 immunoprecipitated with an anti-BRCA1 antibody was analyzed by immunoblotting with an anti-Ub antibody. Quantification of ubiquitylated BRCA1 and the blot with long exposure are shown in Supplementary Fig. 10b and d. **d, e** The data are representative of three independent experiments. Source data are provided as a Source Data file.

interacts with cellular proteins and protects them from Ub-mediated degradation[28,30]. We showed that IARS1 directly interacts with the heterodimeric RING domains of the BRCA1–BARD1 complex and the BRCT domain of BRCA1 through its UBX domain(s) and prevents their ubiquitylation. When anchored on the BRCA1–BARD1 RING domains and/or the BRCT domain of BRCA1, the UBX domains can competitively interact at or near the ubiquitination site of BRCA1 and, together with the unbound catalytic domain, may hinder access of Ub-conjugating enzymes or proteases (Figs. 3a, 4e). Association of ARSs with partners in the MSC is critical for the nuclear localization of ARSs[10]. Removal of IARS1 UBX might affect the localizations of associated proteins (EPRS1, LARS1, and AIMPs) and disable their orthogonal functions. For example, AIMP3 (p18) affects the activities of ATM and ATR during the DNA damage response[53]. Thus, removal of IARS1 might affect ATM activation and associated metabolism during DNA damage signaling.

We showed that IARS1 directly binds to BRCA1 and regulates its ubiquitylation and stability. Since the MSC is assembled in a hierarchical manner such that the stability of each component depends on the other components[7,20] and several MSC components were reported to interact with BRCA1[32], depletion of IARS1 could disrupt the assembly of the MSC and/or decreases the stability of the MSC components, which would subsequently affect the stability of BRCA1. Interestingly, the depletion of two MSC components, AIMP3 and EPRS1 did not affect the BRCA1 level (Supplementary Fig. 7f, g), suggesting the functional specificity of IARS1 in the control of BRCA1 stability. However, we do not exclude the possibility that IARS1 indirectly regulates the stability of BRCA1 via other interacting proteins.

The BRCA1–BARD1 complex interacts with proteins involved in DNA repair, replication fork regulation, and cell cycle regulation[54]. Consistent with the direct interaction of IARS1 with BRCA1, IARS1 depletion reduced the BRCA1 protein level and impaired DNA repair process, such as HR and Rad51 focus formation. IARS1 depletion also significantly reduced the resection of DSB ends comparable with depletion of CtIP. It is unclear whether decreased end resection is a direct effect of BRCA1 destabilization. Although BRCA1 interacts with CtIP and increases the rate of end resection[34,54], there is no evidence that BRCA1 regulates the activity of CtIP[48,49]. Nevertheless, depletion of IARS1 reduced the level of CtIP, a critical endonuclease that promotes DSB end processing, suggesting that IARS1-mediated CtIP stabilization may contribute to end resection.

How is IARS1 released from the complex? Phosphorylation of Ser886 and Ser999 in the WHEP domain releases EPRS1 from the MSC[12]. Proteomics analyses suggest that the two conserved Ser residues and one Tyr residue in the hairpin loop can be phosphorylated[29]. Although the physiological significance of phosphorylation of these residues has not been established, we showed that phosphomimetic mutation of Ser688 and Ser691 released IARS1 from EARS1. Thus, we speculate that phosphorylation of these sites releases IARS1 from the complex in the physiological condition.

BRCA1 directly regulates HR, and we predicted that depletion of BRCA1 would reduce SCE[33,54]. Thus, increased SCE upon depletion of IARS1 observed in our analysis is unlikely to be mediated by BRCA1. Downregulation of several proteins involved in replication fork regulation is associated with increased SCE. For example, BLM helicase and ATR are reported to suppress SCE[55,56]. Furthermore, ATAD5, a PCNA-regulating factor, at replication forks controls SCE and its defect increases SCE[57]. Thus, we speculate that the depletion of IARS1 also affects DNA replication forks in addition to DNA repair. This suggests that the involvement of IARS1 in genomic stability is not limited to BRCA1-mediated DNA repair.

Why have ARSs evolved to regulate DNA metabolism? Because protein synthesis is probably the last checkpoint for genomic stability upon DNA damage[58], it is crucial to check genomic stability at the translation level for proper cell growth and proliferation. Several lines of evidence support the importance of ARSs in genomic stability. For example, AIMP3 regulates p53 activity via activation of ATM and ATR[53]; tryptophanyl-tRNA synthetase forms a complex with DNA-dependent protein kinase and PARP1, and regulates p53 activity[59]; and oxidative stress induces nuclear localization of tyrosyl-tRNA synthetase, which in turn activates the transcription factor E2F1 to upregulate expression of DNA damage repair genes[60]. We further elucidated the function of ARSs in the maintenance of genomic stability by revealing that IARS1 regulates DNA repair. Proteomics analysis suggests that more than 100 proteins are expected to interact with IARS1[31]. Thus, we speculate that the role of IARS1 UBX domains may not be limited to the maintenance of genomic stability and DNA repair factors.

## Methods
### Cloning, expression, and purification

For biochemical analyses, genes encoding full-length human IARS1 (residues 1–1262) and C-terminal-truncated IARS1 (residues 1–1150 and 1–1081) were amplified by PCR and inserted into the pFASTbacHTb vector with a His6 tag at the N-terminus. All constructs were expressed in Sf9 cells (Life Technologies) at 27 °C for 72 h. The proteins were purified using a Ni$^{2+}$ column, an ion-exchange (IEX) column, and size exclusion chromatography (SEC). To purify the LARS1 UNE-L domain (residues 1062–1176), the corresponding gene was inserted into the pGEX6P1 vector and expressed in *Escherichia coli* Rosetta (DE3) cells. The UNE-L domain was initially purified by GST affinity chromatography. After treatment with PreScission protease (GE Healthcare), the protein was further purified by IEX and SEC.

To generate human *eprs1* constructs, three versions of genes (residues 1–706, residues 1–1015, and Δ684–694) were amplified, inserted into the pGEX4T3 vector, expressed in *E. coli* Rosetta (DE3) cells, and purified using a GST column followed by gel filtration. For the pull-down assay, *Gg*EARS1 catalytic domain mutants (residues 176–706 with Ser688Glu, Ser691Glu, and Ser688Glu/Ser691Glu, and Δ684–694) were inserted into the pET28a vector, expressed in *E. coli* Rosetta (DE3) cells, and purified by His-affinity chromatography, IEX, and SEC.

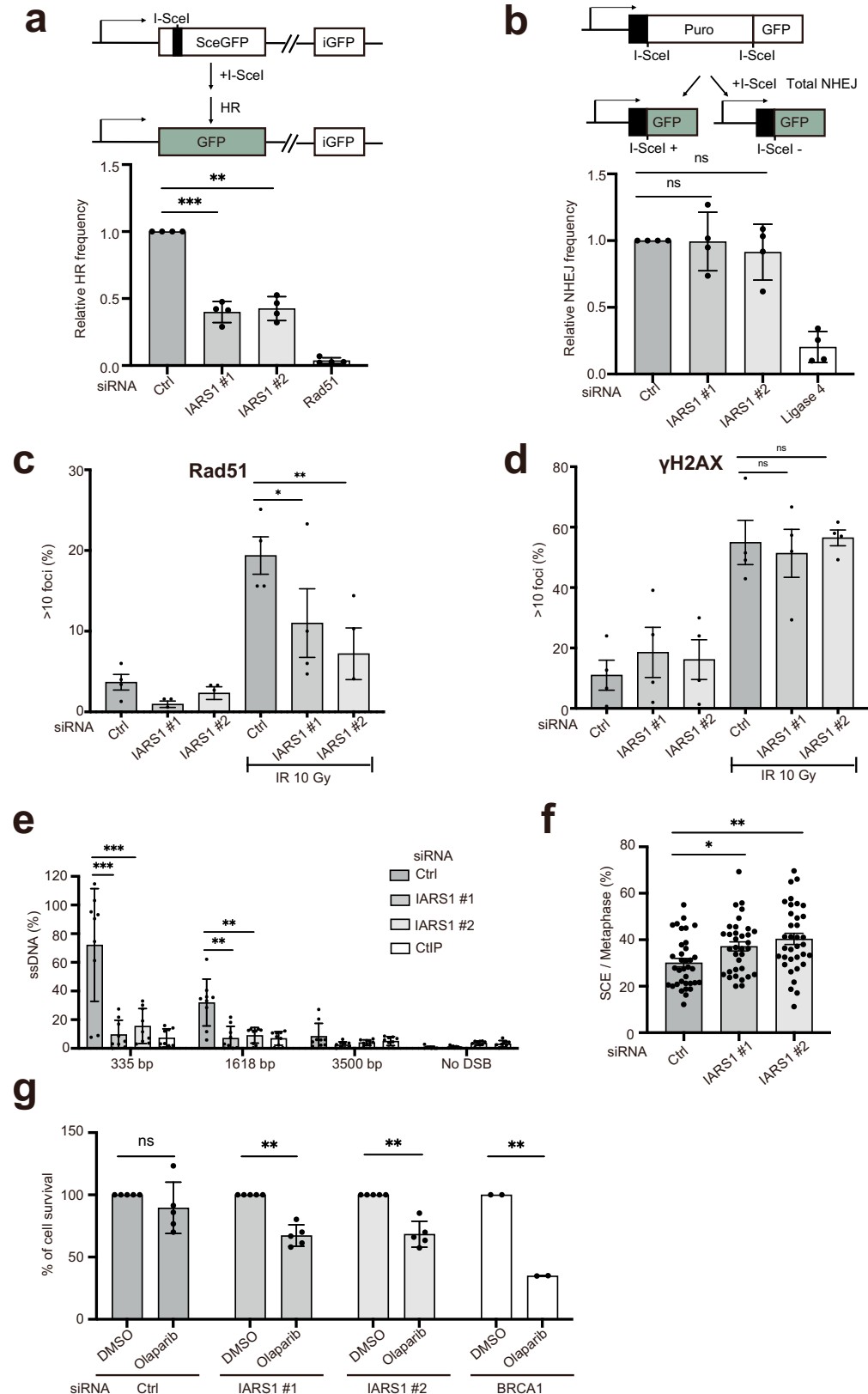

## Affinity pull-down assay

Human His-IARS1 or IARS1ΔUBX2 protein was mixed with human GST-EARS1 or EARS1Δhairpin (residues 1−1015 and Δ684−694) proteins at a ratio of 1:1.5, respectively, and incubated for 30 min at 4 °C. The protein mixtures were loaded onto Ni-sepharose beads for 2 hr at 4 °C and eluted with 250 mM imidazole prepared in His A buffer (20 mM Tris-HCl, pH 7.4, 300 mM NaCl, 7 mM 2-mercaptoethanol, and 5% glycerol) for further analysis. A His-tagged *Gg*EARS1 phosphomimetic mutant (S688E, S691E, or S688E/S691E) or the hairpin loop-deleted mutant was mixed with the GST-UNE-I domain at a ratio of 1:1.5, respectively, and incubated for 30 min at 4 °C. The protein mixtures were pulled down as in the IARS1-EARS1 pull-down assay. For GST pull-down

**Fig. 5 | Downregulation of BRCA1 upon IARS1 depletion leads to impaired DNA repair. a** Schematic of the DR-GFP reporter assay (top). I-SceI-DSB-induced HR efficiencies were determined in DR-U2OS cells upon treatment with control siRNA, two siRNA$^{IARS1}$, or siRNA$^{Rad51}$ (bottom). **b** Schematic of the EJ5-GFP reporter assay (top). The efficiency of NHEJ in EJ5-U2OS cells was measured after transfection of control, two siRNA$^{IARS1}$, or siRNA$^{Ligase4}$ (bottom). Data are represented as mean values ± SD ($n = 4$, independent experiments) in **a** and **b**. **$P < 0.01$, ***$P < 0.001$, ns, not significant by the analysis of two-tailed paired Student's t-test. **c** Quantification of Rad51 foci in U2OS cells treated with control siRNA or two siRNA$^{IARS1}$ after exposure to 10 Gy ionizing radiation (IR) or sham irradiation. **d** Quantification of γH2AX in nuclei of U2OS cells treated with control siRNA or two siRNA$^{IARS1}$ after exposure to 10 Gy IR or sham irradiation. The mean values ± standard error of the mean of four (siRNA Ctrl and siRNA$^{IARS1}$#1 and #2) independent experiments are shown in **c** and **d**. *$P < 0.05$, **$P < 0.01$, ns, not significant by the analysis of two-tailed paired Student's t-test. **e** Quantification of resected ssDNA at DSB sites using the ER-AsiSI

system upon treatment with control siRNA, two siRNA$^{IARS1}$ or siRNA$^{CtIP}$. After 4-OHT treatment for 4 hr to induce DSBs, resected DNA was quantified by qPCR at the indicated nucleotides from AsiSI-induced DSBs. The mean values ± SD of three (siRNA Ctrl, siRNA$^{IARS1}$#1 and #2, and CtIP) independent experiments are shown. **$P < 0.01$, ***$P < 0.001$ by the analysis of two-tailed paired Student's t-test. **f** SCE analysis in HeLa cells treated with control siRNA or two siRNA$^{IARS1}$. The mean values ± SD of SCE/Total metaphase chromosome ($n = 35$ cells) are shown. *$P < 0.05$, **$P < 0.01$ by the analysis of two-tailed paired Student's t-test. **g** Clonogenic survival of HeLa cells upon treatment with control siRNA, two siRNA$^{IARS1}$, or siRNA$^{BRCA1}$ was measured after continuous exposure to olaparib for about 2 weeks. The mean values ± SD of five (siRNA Ctrl and siRNA$^{IARS1}$#1 and #2) or two (siRNA$^{BRCA1}$) independent experiments are shown. **$P < 0.01$, ns, not significant by the analysis of two-tailed paired Student's t-test. **a–g** Source data are provided as a Source Data file.

analysis, samples eluted from the Ni-sepharose beads with or without GgIARS1 were diluted 2-fold in His A buffer and then incubated with GST-sepharose beads for 2 hr at 4 °C. The beads were washed three times with His A buffer and eluted with a buffer containing 10 mM L-glutathione. All eluted proteins were analyzed by SDS-PAGE and stained with Coomassie Blue.

## Crystallization of the EARS1–IARS1 complex
For crystallization, genes encoding the GgEARS1 catalytic domain (residues 176–706) and UNE-I domains of GgIARS1 (residues 965–1265) were fused by a 12 residue linker (SGSLVPRGSSGS), which contained a thrombin cleavage site, inserted into the pET28a vector (Novagen), and expressed in *E. coli* Rosetta (DE3) cells. The EARS1–UNE-I chimera was initially purified using a Ni$^{2+}$ column, and the intervening linker was cleaved by the thrombin protease during dialysis. The EARS1–IARS1 complex protein was trimmed by subtilisin and further purified by IEX and SEC. The final purified complex protein was concentrated to 30 mg/mL and used for crystallization. GgEARS1–IARS1 complex microcrystals were obtained using the batch crystallization method. A solution containing GgEARS1–IARS1 complex protein and crystallization solution (0.84 M NaH$_2$PO$_4$, 1.76 M K$_2$HPO$_4$, and 0.1 M sodium acetate, pH 4.5) were mixed at a ratio of 1:2, respectively, and incubated for 1 week at 20 °C.

## Determination of the EARS1–IARS1 complex structure
Diffraction data for GgEARS1–IARS1 microcrystals were collected using the fixed-target serial femtosecond crystallography method with the PAL-XFEL[61]. Data collection and processing are described in Supplementary Information. The structure of the GgEARS1–IARS1 complex was determined by the molecular replacement method. The crystal structure of GgEARS1 alone was initially determined at 2.5 Å resolution and used to locate the catalytic domains of GgEARS1 molecules in the GgEARS1–IARS1 complex (Supplementary information). The catalytic domains of GgEARS1 molecules were located with the PHASER[62] program and then a search for the anticodon-binding domain of GgEARS1 was performed. After density modification, an electron density map was generated at 2.4 Å resolution using the PHENIX[63] program and was of good quality, which allowed protein molecules to be built. Successive rounds of manual building with COOT[64] and refinement with PHENIX (using rigid body refinement, xyz coordinates, and individual B-factors) were employed to build the complete model. The final refined model of the GgEARS1-IARS1 complex at 2.4 Å resolution ($R_{work}/R_{free}$ of 19.3%/24.3%) contained 94.4% of residues in its most favored region and 0.6% residues in the disallowed region (Supplementary Table 1). The model did not include residues 1007, 1094, 1166, and 1167 of the IARS1 UNE-I domain. Figures were generated using Pymol Molecular Graphics System version 2.3 (Schrödinger).

## Immunoprecipitation
After being transfected for 48 h, cells were washed twice with ice-cold phosphate-buffered saline and lysed with IP buffer (50 mM Tris pH 7.5, 150 mM NaCl, 1% Triton X-100, and 1 mM EDTA) or co-IP buffer (50 mM Tris pH 7.5, 150 mM NaCl, 1% Triton X-100, and 10% glycerol) supplemented with 1 mM phenylmethylsulfonyl fluoride and a protease inhibitor cocktail (Roche). The soluble fraction of the cell lysate containing Flag-tagged EPRS1 proteins was isolated by centrifugation and incubated with anti-Flag M2 agarose affinity beads (Sigma) for 2 h at 4 °C. The beads were washed three times with IP buffer and eluted with IP buffer containing 0.4 mg/mL Flag peptide. The isolated soluble fraction of the cell lysate containing Flag-tagged IARS1 proteins was incubated with anti-Flag G1 resin (Genscript) overnight at 4 °C. The beads were washed three times with co-IP buffer and proteins were eluted with 2 × SDS loading buffer. Co-immunoprecipitated proteins were analyzed by immunoblotting.

For BRCA1 immunoprecipitation, an anti-BRCA1 antibody (D-9, Santa Cruz) was mixed with the cell lysate overnight at 4 °C with constant rotation. Protein G beads (GE Healthcare) were added and incubated for 2 hr. After four washes with 1/2 buffer X, precipitates were dissolved in 2 × SDS loading buffer and separated by SDS-PAGE. Cytoplasmic and nuclear fractions were prepared using NE-PER Nuclear and Cytoplasmic Extraction Reagents (Thermo Scientific). The isolated nuclear fraction was incubated with agarose conjugated with an anti-BRCA1 antibody (D-9 AC, Santa Cruz) overnight at 4 °C with constant rotation. After five washes with 1/2 buffer X, precipitates were dissolved in 2× SDS loading buffer and separated by SDS-PAGE.

## Affinity purification-mass spectrometry analysis
HEK293T cells transiently expressing Flag-tagged IARS1 were lysed with buffer X, and the crude lysates were incubated with Flag-M2 beads. Bead-bound proteins were washed and eluted with 0.4 mg/mL Flag peptide in buffer X. Proteins were separated by SDS-PAGE and visualized by Coomassie Blue staining. For liquid chromatography-tandem mass spectrometry analyses, gel lanes were sliced into bands and processed as follows. Briefly, the protein bands were divided into 10-mm sections and in-gel digestion was performed with trypsin. Trypsin digest products were separated by online reversed-phase chromatography using a Thermo Scientific EASY-nLC 1200 ultra-high performance liquid chromatography system equipped with an autosampler, an Acclaim PepMap™ 100 reversed-phase peptide trap (75 μm inner diameter and 2 cm length), and a PepMap™ RSLC C18 reversed-phase analytical column (75 μm inner diameter, 15 cm length, and 3 μm particle size). Samples were then subjected to electrospray ionization at a flow rate of 300 nL/min. The chromatography system was coupled in-line with an Orbitrap Fusion Lumos Mass Spectrometer. Obtained spectra were screened against the UniProt human database using Proteome Discoverer Sorcerer 2.1 software with a SEQUEST-based

search algorithm. The comparative analysis of proteins identified in this study was performed using Scaffold 4 Q + S.

## Clonogenic assays

HeLa cells were transfected with scrambled or IARS1-targeting siRNA using Lipofectamine RNAiMAX. At 24 hr after transfection, cells were trypsinized and re-plated in 6-well plates at a density of $1 \times 10^4$ cells per well in 2 mL. Cells were treated with olaparib (PARP inhibitor) at 37 °C for 14 days. For continuous exposure to the inhibitor, cells were re-fed with fresh medium containing the inhibitor every 4 days. Cells were stained with 0.5% crystal violet before colonies were counted. Clonogenic survival was determined by dividing the number of colonies on each treated plate by the number of colonies on the untreated plate.

## I-SceI-induced DSB assay

To measure DNA repair efficiencies, transfected cells were plated in a 12-well plate at a density of $1 \times 10^5$ cells per well after 2 days. The following day, cells were co-transfected with 0.5 μg of the I-SceI expression vector or empty vector and 0.1 μg of the dsRED vector (used as a transfection control) in 0.1 mL Opti-MEM containing 3 μL Lipofectamine 3000 (Invitrogen). After 6 hr, the medium was removed and replaced with growth medium. At 2 days after I-SceI transfection, the percentage of GFP + cells was analyzed using a Becton Dickinson FACSVerse flow cytometer. The DNA repair efficiencies were determined as described previously[65]. Experiments were repeated at least three times, and the average values were used.

## Measurement of resection in mammalian cells

The level of resection adjacent to specific DSBs was measured by quantitative polymerase chain reaction (qPCR). The sequences of qPCR primers and probes are shown in Supplementary Table 2. Thirty-six microliters of a genomic DNA sample were digested or mock-digested with 80 units of restriction enzymes (BsrGI and HindIII-HF, New England Biolabs) overnight at 37 °C. Three microliters of the digested or mock-digested sample were used as templates in a qPCR of 25 μL containing 12.5 μL TaqMan Universal PCR Master Mix (ABI), 0.5 mM of each primer, and 0.2 mM probe using a QuantStudio 7 Flex Real-Time PCR System (ABI). The percentage of ssDNA (ssDNA%) generated by resection at selected sites was determined as previously described[66]. Briefly, for each sample, a Ct was calculated by subtracting the Ct value of the mock-digested sample from the Ct value of the digested sample. ssDNA% was calculated using the following equation: ssDNA% = $1/(2^{(4Ct-1)}+0.5)*100$[67].

## SCE assay

Cells were cultured in medium containing BrdU at a final concentration of 25 μg/mL for 48 hr. Colcemid (0.2 μg/mL) was added for the final 4 hr. Then, metaphase cells were harvested by trypsinization, swollen in 0.075 M KCl for 15 min at 37 °C, fixed twice with methanol:acetic acid (3:1), dropped onto glass microscope slides, and stained with 5% Giemsa. Images were acquired using a fluorescence microscope (BX53, Olympus). At least 35 metaphase cells were randomly analyzed in each condition.

The following materials and methods are described in Supplementary Information: Crystallization of EARS1 and data collection, Determination of the EARS1 structure, Data collection using XFELs, SFX data processing, Cell culture, cDNAs, siRNAs, Antibodies, Transfection of cell lines, Immunoblot analysis, RNA extraction and cDNA synthesis, Quantitative real-time PCR, Immunofluorescence microscopy, and Laser microirradiation.

## Reporting summary

Further information on research design is available in the Nature Portfolio Reporting Summary linked to this article.

## Data availability

Atomic coordinates and the structure factors have been deposited in the RCSB Protein data bank (PDB) under the following accession numbers: 7WRS (EARS1-IARS1 complex) and 7WRU (apo EARS1). The mass spectrometry proteomics data have been deposited to the ProteomeXchange Consortium via the PRIDE[68] partner repository with the dataset identifiers PXD031261. Source data are provided with this paper.

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

## Acknowledgements

We thank Y.R. Kim and M.S. Kim (POSTECH), Y. Jung and S.J. Lee (KAIST), M. Wang (UT Health San Antonio), and D. Lee and J. Chung (SNU) for helpful comments and technical help. We thank N.Y. Ha (IBS) for assistance with preparing samples for MS analysis. This work was supported by grants from the National Research Foundation of Korea (NRF) funded by the Korean government (MSIT, No. 2021R1A2C301335711 and 2019M3E5D6066058 to Y.C., No. 2017M3A9F6029736 to Y.C. and W.K.C., No. 2020M3H1A1075314 to S.C., No. 2020R1A2C3013255 and No. 2018R1A5A2025286 to Y.-S.L., No. NRF-2021R1A3B1076605 and Yonsei University 2021-22-0291 to S.K.), the Institute for Basic Science (IBS-R022-D1 to K.J.M.), the BK21 program (Ministry of Education, Y.C.), and Young Investigator Award from Max and Minnie Tomerlin Voelcker Fund, CPRIT RP210102, and NIH R01GM141091 (W.Z.).

## Author contributions

S.C. carried out protein expression, purification, crystallization, and structure determination. D.L., J.P., K.L., and W.K.C. prepared the motion stage for fixed target scanning. K.H.N. prepared the sample loading chip. S.C. and J.K prepared the crystal sample. S.C., J.P., and Y.C. performed the data collection. S.C. and K.H.N. performed data processing. S.C. conducted most of the cell-based experiments with help from M.-S.K. and S.H.H. Y.K., D.S.A., N.-O.Y., and W.Z purified the proteins and performed the in vitro pull-down assay. G.-I.M., B.L., and Y.-S.L. performed the ubiquitylation assay. B.-G.K performed the MS analysis. M.W. and J.-M.O. performed immunostaining and laser microirradiation, respectively. E.A.L. performed the DSB and end resection assays. J.S.R. performed the SCE and clonogenic assays. S.C., K.J.M., and Y.C. designed the research with help from S.H.R., K.-T.K. and S.K. S.C. and Y.C. wrote the manuscript with help from K.J.M.

## Competing interests

The authors declare no competing interests.
