## [Peer Review File · Nature Communications]

Regulation of BRCA1 stability through the tandem UBX domains of isoleucyl-tRNA synthetase 1REVIEWER COMMENTS

Reviewer #1 (Remarks to the Author):

In the course of eukaryote evolution, aminoacyl-tRNA synthetases (aaRSs) acquired unique (UNE) domains which were appended to the catalytic core domains. These UNE domains are often not essential for the aaRS's canonical role in translation but play important non-translational functions. In their manuscript "Regulation of BRCA1 stability through the tandem UBX domains of isoleucyl-tRNA synthetase", Chung et al. provide the first structural and functional characterization of UNE-I, a unique, previously uncharacterized domain appended to vertebrate isoleucyl-tRNA synthetases.

The main result of Chung et al. is the structure of UNE-I from chicken IleRS in complex with the GluRS portion of the bifunctional EPRS. Using this structure as a vantage point, the authors go on to perform a functional analysis of UNE-I (or rather IleRS) in the context of the multisynthetase complex (MSC), as well as in potential extra-translational functions. 1.) They show that UNE-I, which consists of tandem UBX domains, is the primary association site between IleRS and GluRS in the MSC and plays an important role in stabilizing MSC components. 2.) Using phosphomimetic mutations, they claim to show that phosphorylation of GluRS in the hairpin loop interacting with UNE-I allows the regulated release of IleRS from EPRS. 3.) They identify BRCA1 as a potential non-MSC interaction partner for UNE-I and conclude that BRCA1 is protected by IleRS from ubiquitin-mediated degradation, finally leading the authors to propose a critical role for UNE-I in the regulation of DNA repair.

Overall, this manuscript provides an important contribution to the field. As the authors correctly state, IleRS with its UNE-I domain is one of the least studied synthetases and their work represents an important step forward to unravel its function. In particular the structural work is well executed and provides a solid basis for further characterization of UNE-I. However, the data to link UNE-I directly to BRCA1 stability and function in DNA repair is, although interesting, less convincing. On the one hand, it remains unclear why the authors focus on BRCA1. On the other, they do not provide sufficient evidence to show that the observed destabilization of BRCA1 and regulatory effect on DNA repair are a direct function of the UNE-I domain. Moreover, on several other occasions the authors make claims that are not fully supported by their data.

Major points:

1. P. 3 line 49: As one of their main results the authors claim that "Phosphorylation of the two serine residues in the hairpin loop releases IleRS from the complex." Although the authors indeed demonstrate that the introduction of phosphomimetic serine-to-glutamate mutations into putative phosphorylation sites in GluRS disrupts the IleRS-GluRS complex, no evidence is provided to warrant the above statement in the abstract.
2. P. 5 line 107: The claim "Nuclear localized IleRS interacts with BRCA1" is not backed up by any direct data. It should be made clear that this is an extrapolation from the observation that a small portion of IleRS (ca. 2%) is localized to the nucleus and that IleRS interacts with BRCA1 in vitro.
3. P. 3 line 52: The most important claim by the authors is to "show that the IleRS UNE-I domain plays a critical role in maintenance of genomic stability via regulation of DNA repair" and "These results suggest that BRCA1 degradation is primarily regulated by the UBX domains of IleRS..." (P12 line 341). The evidence from this manuscript supporting these claims is ambiguous:
 - a. The authors appear to have chosen BRCA1 more or less randomly from the ca. 280 interactors of full-length IleRS reported in the BioGrid database. 28% of these are, like BRCA1, proteins involved in UBL conjugation pathways and cell cycle regulation (Fig. 3b). How do the authors rule out the possibility that some or all effects of IleRS depletion on BRCA1 ubiquitination and stability are indirect through any of these other potential interaction partners?
 - b. Nearly all experiments that were interpreted as showing a direct effect of UNE-I on BRCA1 stability and function were done by depleting full-length IleRS. The only exception is the data presented in Figure 4b. However, even this data is indirect as it merely shows the absence of a stabilizing effect by IleRS lacking UNE-I. The authors should perform the corresponding experiment overexpressing NLS-

UNE-I to show that UNE-I alone is able to stabilize BRCA1 in the cell. The same is true for their follow-up experiment on the ubiquitination of BRCA1 (Fig. 4d).

c. Given the importance of IleRS for the stability of the MSC and its components, how do the authors exclude the possibility that some or even all the effects observed by IleRS depletion are indirect through the simultaneous destabilization of the MSC, destabilization of interacting MSC components, or the release of other MSC components such as AIMP3?

d. P14 line 400: In this context, it is interesting that the authors themselves speculate that "Protection of the MSC via the UBX domains may prevent abortive functions in various cellular metabolism." Given that nearly all MSC components are reported to interact with BRCA1 (Ertych et al. 2016 PNAS <https://doi.org/10.1073/pnas.1525129113>), how do the authors rule out the possibility that their IleRS-depletion results are due to an indirect effect through destabilizing the MSC, thus inducing "abortive functions" in other cellular pathways?

In addition to overexpressing stand-alone UNE-I to demonstrate its ability to compensate the depletion of IleRS (see point #3b), the authors could deplete another MSC-associated aaRS (e.g. EPRS) as a control to demonstrate specificity of the observed response to IleRS.

e. The Co-IP to show direct interaction of BRCA1 with UNE-I but not with the catalytic portion of IleRS (Fig. 3e) is not clean. The band for UNE-I is very weak and the region where the truncated IleRS might migrate is covered by a smear. Given the importance of this experiment for the central conclusions of this work, it may be important to repeat it.

Minor:

1. Figure 2b is very busy and may benefit from removing some information. E.g. the sticks for residues that are not labeled.
2. P. 8: In the section "IleRS binds to the GluRS...", the first and third paragraphs are partially redundant and could be combined for simplification.
3. P11 line 311: "Three siRNAs targeted the mRNA and one targeted the 3'UTR" should be "targeted the coding region of the mRNA and one..."?
4. P. 11 line 315-316: Should be "did not affect the mRNA level of BRCA1"?
5. P 13 line 392: How do the authors know that "its [UNE-I] primary function is to interact with cellular proteins and protect them from degradation"? Since the authors focus exclusively on this aspect of UNE-I function without excluding any others, it is unclear how they reach to this statement.

Reviewer #2 (Remarks to the Author):

Chung et. al., present a novel model whereby isoleucyl-tRNA synthetase (IleRS) interacts with BRCA1 and protects it from ubiquitin-dependent degradation. First, the authors solve the structure of the non-catalytic UNE-I of IleRS in complex with a multi-synthetase complex interactor, GluRS. The authors identify two ubiquitin-like UBX domains within the C-terminus of IleRS, which they propose functions by prohibiting ubiquitination of various IleRS partners, thereby preventing their proteolytic degradation. Specifically, IleRS was found to interact with the tumor suppressor BRCA1, and depletion of IleRS resulted in decreased BRCA1 protein levels, reduced homologous recombination efficiency, and PARP inhibitor sensitivity. The authors claim that BRCA1 ubiquitination is more prominent after depletion of IleRS, which corresponds to decreased BRCA1. This is supported by BRCA1 levels remaining high in the presence of the proteasome inhibitor MG132.

Although the proposed model is interesting, it is preliminary as some of the data are not convincing.

Specific Comments:

1. Some of the pulldown data are difficult to interpret. For example, the lack of a lane showing the last wash in Fig. 1b is a concern. Similarly, the protein preps have several other species present, and in the case of His-IleRS(1-1262), there is a band that runs similar to the size of GST-GluRS(1-1050). Cleaner preps or a blot would increase the confidence that the interactions are real.
2. The authors use the data from Fig. 3f to conclude that IleRS interacts with the RING and BRCT domains of BRCA1, but IleRS interacts with BRCA1 when either domain is absent, suggesting these domains are not involved in the interaction. The interaction predictions are thus also in question. Does IleRS interact with isolated RING or BRCT domains of BRCA1 or BARD1?
3. In lines 330-331, the authors claim WT IleRS restores BRCA1 levels, but the construct lacking UBX domain did not. This conclusion is debatable as the BRCA1 levels in the WT IleRS lane are very comparable to the EV lane.
4. Quantification of BRCA1 ubiquitination under various conditions (Fig. 4d) would go a long way towards supporting the model.
5. Why do the authors use siRNAIleRS #1 and #2 (are the labels correct?) for the HR and NHEJ assays in Fig. 5? Since these siRNAs reduced BRCA1 mRNA levels (Supplemental Fig. 7), changes in HR efficiency aren't necessarily a function of proteolytically protected BRCA1. Is the same phenotype observed with the other two siRNAs used in previous experiments?
6. The manuscript needs a thorough editing. There are numerous unclear phrasings and grammatical errors.
7. Line 121 refers to Fig. 1a in the context of describing the GluProRS chimera, but this seems like an incorrect reference.

Minor Comments:

1. "UNE-I" is written as "UNE-L" on line 129.
2. The word "weakly" is used to describe the interactions between IleRS some MSC components based on mass spec results (see line 116), but the affinity of these interactions cannot be concluded from this type of experiment. For example, the interaction may be "strong," but only occurs under specific contexts. Consider rewording.
3. Why did the authors purify and digest a fusion protein instead of co-expressing the fragments?
4. Does knockdown of IleRS (presumably essential?) cause any cell-cycle or general growth defects?
5. Why are BRCA1 levels inconsistent in Fig. 3e?
6. Use consistent naming when referring to the non-UNE-1 domain of IleRS instead of changing between "main," "catalytic," or "canonical." Matching with the naming in Fig. 1c is ideal.
7. Is BARD1 also immunoprecipitated with IleRS?
8. The claim that IleRS knockdown reduces EXO1 levels (Supplemental Fig. 11a) is not convincing.
9. EXO1 is sometimes written as "Exo1" or "EXO1." It should be consistently "EXO1."

Reviewer #3 (Remarks to the Author):

Aminoacyl-tRNA synthetases (aaRS) have a wide array of functions in the cell, in addition to their role in charging tRNAs with their cognate amino acids for translation. To this end, most aaRS enzymes have additional unique domains (UNEs) that allow them to mediate their diverse cellular functions.

In this study, the authors investigate different functional aspects of IleRS, a relatively uncharacterized enzyme of the multi-synthetase complex (MSC). Using a combination of structural, biochemical and cell biology experiments, the authors delineate the interaction of IleRS with another aaRS, the GluRS, as well as its interactions with other cellular factors such as BRCA1. The UNE-I domain of IleRS comprises two UBX domains and one KH domain. The authors determined the X-ray crystal structure of the IleRS-UNE-I region in complex with GluRS at fairly high resolution. The two UBX domains interact extensively with the GluRS C-terminus (including its helical subdomain and the anticodon binding domains). The structural observations led the authors to investigate if the UBX domains of

IleRS interact with a diverse array of cellular factors to protect them from ubiquitination and proteasomal degradation, and if so what the impact of this interaction on cellular function might be. They found that the protein BRCA1 is stabilized by interaction with IleRS and absence of this interaction has consequences on DNA damage repair, particularly homology repair.

The structural work presented in this study is strong and is well described. The X-ray crystal structure of the GluRS-IleRS complex gives the reader an insight into interactions in the MSC. The follow-up biochemical and cell biology work presented in the manuscript is however less convincing. A few points of concern are noted below:

- Page 5, lines 126-130: the authors claim formation of a ternary complex between GluRS, IleRS and LeuRS when no direct evidence for formation of a ternary complex has been presented. What the authors do show are two binary complexes of GluRS-IleRS and IleRS-LeuRS involving distinct regions of IleRS. This statement should either be modified or evidence of a ternary complex should be shown preferably by size-exclusion chromatography.
- Supplementary Figure 7a/b: The authors claim that "In IleRS targeting siRNA-treated cells, the stabilities of LeuRS, USP14, and BRCA1 were significantly decreased, while those of GluProRS and CDK4 were moderately decreased". A careful inspection of these figures shows moderate decrease of BRCA1 and LeuRS and a very slight decrease, if at all of the other proteins. Perhaps showing different exposures or a different blot would help to support the authors' claims.
- The data in Figure 3e and Supplementary figure 8b are unfortunately not convincing at all. The amount of IleRS co-precipitated with BRCA1-BARD1 is negligible and cannot be interpreted as evidence of a complex. Perhaps silver-staining the gel instead of Coomassie staining might work better (just a suggestion). Likewise, the amounts of myc-BRCA1 immunoprecipitated are unequal and one must be very cautious about concluding information about interacting domains from this IP. Since Figure 3e is central to this manuscript, this is a key concern.
- The authors present models of the IleRS-UBX domains bound to RING and BRCT domains of BRCA1. However, as these models are not validated by mutational studies or cross-linking mass-spectrometry, they do not appear to contribute to the overall understanding of the IleRS-BRCA1 interaction. It would be great if the authors could provide some experimental evidence to support these models. If not, it might be better to remove them in the interest of brevity.
- Is it possible to check the nuclear localization of endogenous IleRS by cellular fractionation? Over-expression often leads to change in nucleo-cytoplasmic localization. If not, could the authors compare the levels of the over-expressed protein to the endogenous component?

Responses to reviewers' comments

Responses to reviewer #1's comments

Major points:

1. P. 3 line 49: As one of their main results the authors claim that “**Phosphorylation of the two serine residues in the hairpin loop releases IleRS from the complex.**” Although the authors indeed demonstrate that the introduction of phosphomimetic serine-to-glutamate mutations into putative phosphorylation sites in GluRS disrupts the IleRS-GluRS complex, no evidence is provided to warrant the above statement in the abstract.

>> **P3, line 48**, We agreed with the reviewer and changed the sentence: “**...Phospho-mimetic mutation ..**”. To prove the phosphorylation at these two sites, we attempted to introduce unnatural amino acids at the two sites. However, introducing the double mutation did not efficiently produce the proteins.

2. P. 5 line 107: The claim “**Nuclear localized IleRS interacts with BRCA1**” is not backed up by any direct data. It should be made clear that this is an extrapolation from the observation that a small portion of IleRS (ca. 2%) is localized to the nucleus and that IleRS interacts with BRCA1 in vitro.

>> Please see **Fig 3h, P11, line 297-300**. To prove the interaction between the endogenous IleRS and BRCA1 in nucleus, we isolated the nucleus and performed co-IP analysis using BRCA1-specific antibody in the nuclear extract. As shown in Fig 3h, we demonstrated that **nuclear localized endogenous IleRS interacts with BRCA1**.

3. P. 3 line 52: The most important claim by the authors is to “show that the IleRS UNE-I domain plays a critical role in maintenance of genomic stability via regulation of DNA repair” and “These results suggest that **BRCA1 degradation is primarily regulated by the UBX domains of IleRS...**” (P12 line 341). The evidence from this manuscript supporting these claims is ambiguous:

a. The authors appear to have chosen BRCA1 more or less randomly from the ca. 280 interactors of full-length IleRS reported in the BioGrid database. 28% of these are, like BRCA1, proteins involved in UBL conjugation pathways and cell cycle regulation (Fig. 3b). How do the authors rule out the possibility that **some or all effects of IleRS depletion on BRCA1 ubiquitination and stability are indirect through any of these other potential interaction partners?**

>> As a reviewer points out, **we cannot rule out the possibility** that the effect of IleRS on BRCA1 is through the interaction of other associate proteins. To demonstrate IleRS directly involved in binding to BRCA1 and regulates its ubiquitylation and stability, we did several approaches. In addition to show the direct interaction between the purified BRCA1 (RING and BRCT domains), we showed that overexpression of IleRS UBX domain alone regulates the ubiquitylation of BRCA1 and thereby contribute to the stability of BRCA1 (**Fig 4b, e**). We also

showed that depletion of full-length IleRS increases the ubiquitylation of BRCA1 and decrease its stability (**Fig 4d**). Although it was not possible to test all the interacting proteins, we attempted to test the effect of other MSC components- **AIMP3 and EPRS (Supple Fig. 7f, g, P15, line 440-442)**. None of these affected the level of BRCA1, which suggests that at least these IleRS-associated proteins do not involve in regulating the BRCA1 stability. Despite all these, we cannot exclude a possibility of indirect role of IleRS, so we clearly describe the possible indirect role of IleRS in regulating BRCA1 in the text (**P15, line 437-440**).

b. Nearly all experiments that were interpreted as showing a direct effect of UNE-I on BRCA1 stability and function were done by depleting full-length IleRS. The only exception is the data presented in Figure 4b. However, even this data is indirect as it merely shows the absence of a stabilizing effect by IleRS lacking UNE-I. The authors **should perform the corresponding experiment overexpressing NLS-UNE-I to show that UNE-I alone is able to stabilize BRCA1 in the cell. The same is true for their follow-up experiment on the ubiquitination of BRCA1** (Fig. 4d).

>> Please see the revised **Fig. 4b, e and Supple Fig. 10b**. We **overexpressed UNE-I alone** and confirmed that overexpression of UNE-I alone decreased ubiquitylation of BRCA1 and increased stability of BRCA1. Ubiquitylation and stability of BRCA1 upon overexpression of IleRS lacking the UNE-I domain were similar to those in the control.

c. Given the importance of IleRS for the stability of the MSC and its components, how do the authors exclude the possibility that some or even all the effects observed by IleRS depletion are indirect **through the simultaneous destabilization of the MSC, destabilization of interacting MSC components, or the release of other MSC components such as AIMP3?**

>> It is possible that effects observed by IleRS depletion are indirect through destabilization of MSC or release of other components. However, to remove such possibility, we used UNE-I domain alone and examined the ubiquitylation and stability of BRCA1. This is because full-length IleRS is required for the stability of MSC as the N-terminal editing, catalytic and UNE-I domains contribute to the assembly of MSC (ref 18 and 20 in the text, and Fig 1 in this work). Removal of catalytic domain is likely to affect the assembly of MSC. UNE-I alone can regulate ubiquitylation and stability of BRCA1 (**Fig. 4b and e**), which suggests that the MSC assembly may not be direct factor for the regulation of BRCA1 stability. Nevertheless, we do not exclude a possibility that the MSC assembly could contribute to the observed effect. At least, we performed what the reviewer has suggested. We knocked down two of the MSC components – AIMP3 and EPRS – and examined the level of BRCA1. In both cases, the level of BRCA1 did not change (**Supple Fig. 7f, g**). We described such possibility in the text (**P15, line 437-440**).

d. P14 line 400: In this context, it is interesting that the authors themselves speculate that “Protection of the MSC via the UBX domains may prevent abortive functions in various cellular metabolism.” Given that nearly all MSC components are reported to interact with BRCA1 (Ertych et al. 2016 PNAS <https://doi.org/10.1073/pnas.1525129113>), how do the authors rule out the possibility that **their IleRS-depletion results are due to an indirect effect through destabilizing the MSC**, thus inducing “abortive functions” in other cellular pathways? In addition to overexpressing stand-alone UNE-I to demonstrate its ability to compensate the depletion of IleRS (see point #3b), the authors could **deplete another MSC-associated aaRS (e.g. EPRS)** as a control to demonstrate specificity of the observed response to IleRS.

>> **Please see the Supple Fig. 7f.** We depleted EPRS and examined the effect. We observed that depletion of EPRS - another MSC-associated aaRS did not affect the level of BRCA1.

e. The Co-IP to show **direct interaction of BRCA1 with UNE-I** but not with the catalytic portion of IleRS (Fig. 3e) is not clean. The band for UNE-I is very weak and the region where the truncated IleRS might migrate is covered by a smear. **Given the importance of this experiment for the central conclusions of this work, it may be important to repeat it.**

>> Please see the **revised Figure 3e.** We repeated the experiments and included a significantly improved figure. We also **examined the in vitro interaction of the isolated RING domain and BRCT (of BRCA1) domain** with IleRS (Fig. 3g).

Minor:

1. Figure 2b is very busy and may benefit from removing some information. E.g. the sticks for residues that are not labeled.

>> We divided the figure into two parts (Fig. 2b, c) and removed unlabeled residues.

2. P. 8: In the section “IleRS binds to the GluRS...”, the first and third paragraphs are partially redundant and could be combined for simplification.

>> Please see the revised P8, we have combined the first and third paragraphs and removed the redundant parts.

3. P11 line 311: “Three siRNAs targeted the mRNA and one targeted the 3’UTR” should be **“targeted the coding region of the mRNA and one...”**?

>> Please see the revised text (P11 line 305). We have corrected the sentence.

4. P. 11 line 315-316: Should be “did not affect the mRNA level of BRCA1”?

>> Please see the revised text (P11 line 312). We have corrected the sentence.

5. P 13 line 392: How do the authors know that **“its [UNE-I] primary function is to interact**

with cellular proteins and protect them from degradation”? Since the authors focus exclusively on this aspect of UNE-I function without excluding any others, it is unclear how they reach to this statement.

>> Please see **P14 line 398**. We have corrected to “**...one of its functions is to ..**”

Responses to reviewer #2's comments

Specific Comments:

1. Some of the pulldown data are difficult to interpret. For example, the lack of a lane showing the last wash in Fig. 1b is a concern. Similarly, the protein preps have several other species present, and in the case of His-IleRS(1-1262), there is a band that runs similar to the size of GST-GluRS(1-1050). Cleaner preps or a blot would increase the confidence that the interactions are real.

>> Please see the **revised Fig. 1b**. We have repeated experiment using proteins with high purity and included the improved figure.

2. The authors use the data from Fig. 3f to conclude that IleRS interacts with the RING and BRCT domains of BRCA1, but IleRS interacts with BRCA1 when either domain is absent, suggesting these domains are not involved in the interaction. The interaction predictions are thus also in question. Does **IleRS interact with isolated RING or BRCT domains of BRCA1 or BARD1?**

>> Please see the revised Fig 3g. We examined the interaction of IleRS with the purified heterodimeric RING domains of BRCA1-RARD1 complex, the BRCT domain of BRCA1, and the BRCT domain of BARD1. We used the heterodimeric RING-RING domains as each RING domain (of BRCA1 or BARD1) alone is unstable in an isolated form. **We showed that IleRS interacts with the RING-RING domain and the BRCT domain of BRCA1. But it did not interact with BRCT of BARD1.** We have removed the model because of its uncertainty.

3. In lines 330-331, the authors **claim WT IleRS restores BRCA1 levels, but the construct lacking UBX domain did not**. This conclusion is debatable as the BRCA1 levels in the WT IleRS lane are very comparable to the EV lane.

>> Please see the revised **Fig 4b, e**. We also overexpressed UNE-I domain alone and examined its effect in ubiquitylation and stability of BRCA1.

4. Quantification of BRCA1 ubiquitination under various conditions (Fig. 4d) would go a long way towards supporting the model.

>> We have quantified the data in **Fig 4d, e** and included in **Supple Fig. 10a, b**.

5. Why do the authors use siRNAIleRS #1 and #2 (are the labels correct?) for the HR and NHEJ assays in Fig. 5? Since these siRNAs reduced BRCA1 mRNA levels (Supplemental Fig. 7), changes in HR efficiency aren't necessarily a function of proteolytically protected BRCA1. Is the same phenotype observed with the other two siRNAs used in previous experiments?

>> Please see **Supple Fig. 11a, b**. We performed NHEJ assay using other siRNAs. We have examined four siRNAs and both siRNA#3 and 3'UTR IleRS showed decreased HR efficiency.

6. The manuscript needs a thorough editing. There are numerous unclear phrasings and grammatical errors.

>> We have checked the text and corrected the grammatical error.

7. Line 121 refers to Fig. 1a in the context of describing the GluProRS chimera, but this seems like an incorrect reference.

>> We have deleted Fig 1a reference.

Minor Comments:

1. "UNE-I" is written as "UNE-L" on line 129.

>> UNE-L is correct. Please see the revised sentence (line 118-119, and Supplementary Fig 1a).

2. The word "weakly" is used to describe the interactions between IleRS some MSC components based on mass spec results (see line 116), but the affinity of these interactions cannot be concluded from this type of experiment. For example, the interaction may be "strong," but only occurs under specific contexts. Consider rewording.

>> Please see the revised text (**P. 5 line 113 ~ 117**). We have corrected the paragraph.

3. Why did the authors purify and digest a fusion protein instead of co-expressing the fragments?

>> In the co-expression of the two proteins, we observed a contaminant which we could not remove and therefore interfere crystallization. We included this content in the text (**P. 6 line 137 ~ 139**).

4. Does knockdown of IleRS (presumably essential?) cause any cell-cycle or general growth defects?

>> In “Biallelic *IARS* Mutations Cause Growth Retardation with Prenatal Onset, Intellectual Disability, Muscular Hypotonia, and Infantile Hepatopathy” Am J Hum Genet, (2016) doi: 10.1016/j.ajhg.2016.05.027, Kopajtich et al reported that yeast with nonsense IleRS did not grow properly and Zebrafish with depleted IleRS exhibited a growth defect.

5. Why are BRCA1 levels inconsistent in Fig. 3e?

>> We have repeated this experiment and included a revised Fig. 3e.

6. Use consistent naming when referring to the non-UNE-1 domain of IleRS instead of changing between “main,” “catalytic,” or “canonical.” Matching with the naming in Fig. 1c is ideal.

>> We use the word “catalytic” throughout the text.

7. Is BARD1 also immunoprecipitated with IleRS?

>> **We showed that the heterodimeric RING-RING domains of the BRCA1-BARD1 complex interact with IleRS, but BRCT domain of BARD1 does not interact with IleRS (Fig. 3f). But we do not know whether the RING domain of BARD1 binds to IleRS.** This is because the BARD1 RING domain alone is not stable and must form a heterodimeric RING complex with that of BRCA1 *in vitro*. We could not perform co-IP analysis using isolated RING domain of BARD1 or full length BARD1 alone.

8. The claim that IleRS knockdown reduces EXO1 levels (Supplemental Fig. 11a) is not convincing. !!

>> We have repeated this experiment and included a revised **Supple Fig. 11c**.

9. EXO1 is sometimes written as “Exo1” or “EXO1.” It should be consistently “EXO1.”

>> We have changed “Exo1” to “EXO1”

Responses to reviewer #3's comments

1. Page 5, lines 126-130: the authors claim formation of a ternary complex between GluRS, IleRS and LeuRS when no direct evidence for formation of a ternary complex has been presented. What the authors do show are two binary complexes of GluRS-IleRS and IleRS-LeuRS involving distinct regions of IleRS. This statement should either be modified or evidence of a ternary complex should be shown preferably by size-exclusion chromatography.

>> **Please see Supple Fig 1a, b and P5 line 124-128.** We showed the formation of the IleRS-GluRS-LeuRS ternary complex using a size exclusion chromatography and included a diagram in **Supple Fig. 1b.**

2. Supplementary Figure 7a/b: The authors claim that “In IleRS targeting siRNA-treated cells, the stabilities of LeuRS, USP14, and BRCA1 were significantly decreased, while those of GluProRS and CDK4 were moderately decreased”. A careful inspection of these figures shows moderate decrease of BRCA1 and LeuRS and a very slight decrease, if at all of the other proteins. Perhaps showing different exposures or a different blot would help to support the authors' claims.

>> Please see Supple Fig. 7a and **P. 9 line 256-259.**

3. The data in Figure 3e and Supplementary figure 8b are unfortunately not convincing at all. The amount of IleRS co-precipitated with BRCA1-BARD1 is negligible and cannot be interpreted as evidence of a complex. Perhaps silver-staining the gel instead of Coomassie staining might work better (just a suggestion). Likewise, the amounts of myc-BRCA1 immunoprecipitated are unequal and one must be very cautious about concluding information about interacting domains from this IP. Since Figure 3e is central to this manuscript, this is a key concern.

>> We repeated coIP analysis of IleRS with BRCA1-BARD1 and displayed much **improved data in Fig. 3e.** Furthermore, we showed direct interaction between BRCA1-BARD1 and IleRS (**Fig. 3d**) and between the RING of BRCA1-BARD and IleRS and between the BRCA1 BRCT and IleRS (**Fig. 3f**). We have removed previous Supple Fig. 8.

4. The authors present models of the IleRS-UBX domains bound to RING and BRCT domains of BRCA1. However, as these models are not validated by mutational studies or cross-linking mass-spectrometry, they do not appear to contribute to the overall understanding of the IleRS-BRCA1 interaction. It would be great if the authors could provide some experimental evidence to support these models. If not, it might be better to remove them in the interest of brevity.

>> Although we showed that the RING and BRCT domains of BRCA1 interact with IleRS (Fig. 3f), our data is not sufficient to provide an atomic model for the BRCA1-IleRS interactions. Thus, **we removed the IleRS-BRCA1 models** as suggested by a reviewer.

5. Is it possible to check the nuclear localization of endogenous IleRS by cellular fractionation?

Over-expression often leads to change in nucleo-cytoplasmic localization. If not, could the authors compare the levels of the over-expressed protein to the endogenous component?

>> Please see **revised Fig 3h**. We performed binding analysis using endogenous IleRS using BRCA1 in the nucleus and thereby showing the nuclear localization and interaction with BRCA1.

REVIEWER COMMENTS

Reviewer #1 (Remarks to the Author):

Chung et al. 2nd revision

All my original points and concerns were addressed by the authors in their revised manuscript and point-by-point response. Apart from an improved and more balanced discussion, they also provide important additional data to support their original claims. Unfortunately, there remain some unresolved questions that, in part, arise from the newly presented results.

The most important issue arises from the newly added data in Supplementary Figures 7g and 7f: As suggested, the authors use knocked down EPRS and AIMP3 as a control. They find that neither affects BRCA1 stability. The authors need to include IleRS as a control to show that it remains unaffected as well. This, however, would be inconsistent with earlier data from the one of the coauthors: As demonstrated by Han et al. (J Biol Chem. 2006. doi: 10.1074/jbc.M605211200) knock-down of AIMP3, but especially EPRS, significantly destabilizes and thus reduces levels of IleRS in the cell to a degree similar or even greater than what the authors achieve by direct knock-down of IleRS. However, if EPRS (and AIMP3) knock-down reduces IleRS levels, it should reduce BRCA1 levels indirectly, according to the authors' hypothesis. On the other hand, if IleRS levels indeed remain unaffected by EPRS or AIMP3 knock-down, then the authors need to explain why there is no increase in BRCA1 levels, since the loss of EPRS as an anchor point for IleRS in the MSC should lead to higher levels of "released" IleRS, free to shuttle to the nucleus and interact with BRCA1. These potential inconsistencies need to be clarified.

Another issue arises from the interpretation of the newly added data and the lack of controls for isolated UNE-I interacting with and stabilizing BRCA1 (Fig. 4b, e): In their revised discussion (lines 443-447) and point-by-point response (p. 2), the authors claim that the stabilizing effect of overexpressed UNE-I on BRCA1 must be direct, which should rule out an indirect effect by destabilizing or altering the MSC composition. However, as the authors show in Figure 1a, overexpressed UNE-I efficiently interacts with EPRS in the cell, suggesting that it competes with full-length IleRS for incorporation into the MSC. Thus, full-length IleRS would be released from the MSC, possibly together with other components (as the authors point out: "...full-length IleRS is required for the stability of MSC as the N-terminal editing, catalytic and UNE-I domains contribute to the assembly of MSC". Since the knock-down of endogenous IleRS seems to be inefficient in Figures 4b and e, the observed effects on BRCA1 may still be indirect. Thus, the authors should include controls to show that the positive effect of isolated UNE-I on BRCA1 stability is not due to increased levels of free and nuclear-localized full-length IleRS or due to a general destabilization of the MSC by incorporation of overexpressed UNE-I instead of full-length IleRS.

Figure 3h: The authors should include negative controls in their BRCA1 pull-down, which could ideally include AIMP3 or EPRS, both of which the authors show to be not involved in nuclear stabilization of BRCA1.

Line 116: The authors included "less efficiently" in the sentence "By contrast, the UNE-I domain alone associated most prominently with GluProRS and less efficiently with GlnRS, ArgRS, and AspRS, but did not interact with LeuRS." It is not clear how they reach this conclusion from their mass-spec results, as there are more peptides of AspRS and even LysRS pulled down by UNE-I than EPRS (relative to IleRS). It would be important to know what peptides are listed in Fig. 1a (i.e. absolute numbers or unique peptides), as their number clearly depends on the size of the proteins.

Reviewer #2 (Remarks to the Author):

The revisions are satisfactory and I can now recommend acceptance.

Reviewer #3 (Remarks to the Author):

The authors have satisfactorily addressed all my concerns. The revised manuscript is a great improvement on the original submission and was a pleasure to read. I have only a few minor points that the authors might wish to take note of:

1. Pg. 8, line 212 - "UXB2" should be changed to UBX2.
2. Pg. 9, line 233 - refer to Figure 2c only instead of Figure 2b and 2c.
3. In the legend of Figure 2 - mention that "WCL" is whole cell lysate as some readers may not be familiar with this acronym.

Response to reviewer #1's comments

All my original points and concerns were addressed by the authors in their revised manuscript and point-by-point response. Apart from an improved and more balanced discussion, they also provide important additional data to support their original claims. **Unfortunately, there remain some unresolved questions that, in part, arise from the newly presented results.**

Q1. The most important issue arises from the newly added data in Supplementary Figures 7g and 7f: As suggested, the authors use knocked down EPRS and AIMP3 as a control. They find that neither affects BRCA1 stability. **The authors need to include IleRS as a control to show that it remains unaffected as well.** This, however, would be inconsistent with earlier data from the one of the coauthors: As demonstrated by Han et al. (J Biol Chem. 2006. doi: 10.1074/jbc.M605211200) knock-down of AIMP3, but especially EPRS, significantly destabilizes and thus reduces levels of IleRS in the cell to a degree similar or even greater than what the authors achieve by direct knock-down of IleRS. However, if EPRS (and AIMP3) knock-down reduces IleRS levels, it should reduce BRCA1 levels indirectly, according to the authors' hypothesis. On the other hand, if IleRS levels indeed remain unaffected by EPRS or AIMP3 knock-down, then the authors need to explain why there is no increase in BRCA1 levels, since the loss of EPRS as an anchor point for IleRS in the MSC should lead to higher levels of "released" IleRS, free to shuttle to the nucleus and interact with BRCA1. These potential inconsistencies need to be clarified.

>> Please see revised Supplementary Figures 7g and 7f, in which we included IleRS data. Consistent with Han et al, knock-down of AIMP3 or EPRS decreased the level of IleRS in our analysis. As a reviewer pointed out, this raises a question why decreased IleRS induced by knock-down of AIMP3 or EPRS (or disruption of the MSC assembly) did not affect the level of BRCA1. One possibility is that direct knock-down of IleRS depletes total cellular IleRS (including both of the BRCA1-bound IleRS as well as the MSC-bound IleRS), whereas disruption of the MSC assembly by the suppression of GluProRS or AIMP3 would affect only the MSC-bound IleRS but not the BRCA1-bound IleRS. At least, it appears that reduction of IleRS level at transcription level (by siRNA knockdown) and at post-transcriptional level (by the structural disruption of MSC) would give a different effect on BRCA1 level, implying the additional controlling mechanism to determine the cellular localization of IleRS. Further investigation is needed to address how IleRS can be controlled between the cytosolic MSC and nuclear BRCA1 sites. We are thankful for the reviewer's thoughtful comment.

Besides, we would like to note that the main reason including Supp Fig 7g and f (siRNA data for AIMP3 and IleRS) was to address the reviewer's questions that **".. how do the authors exclude the possibility that some or even all the effects observed by IleRS depletion are indirect through the simultaneous destabilization of the MSC, destabilization of interacting MSC components, or the release of other MSC components such as AIMP3?... how do the authors rule out the possibility that their IleRS-depletion results are due to an indirect effect through destabilizing the MSC, thus inducing "abortive functions" in other cellular pathways?"**

Although we showed that IleRS, but not GluProRS and AIMP3, directly binds BRCA1 to stabilize BRCA1 via the inhibition of BRCA1 ubiquitylation, we do not exclude the possibility that IleRS could still indirectly stabilize BRCA1 through its other interacting factors. We thus

described this possibility in the text (page 15, line 434-446) as shown below.

“We showed that IleRS directly binds to BRCA1 and regulates its ubiquitylation and stability. Since the MSC is assembled in a hierarchical manner such that the stability of each component depends on the other components^{7,20} and several MSC components were reported to interact with BRCA1³², depletion of IleRS could disrupt assembly of the MSC and/or decreases the stability of the MSC components, which would subsequently affect the stability of BRCA1. Interestingly, the depletion of two MSC components, AIMP3 and GluProRS did not affect the BRCA1 level (Supplementary Fig. 7f, g), suggesting the functional specificity of IleRS in the control of BRCA1 stability. However, we do not exclude the possibility that IleRS indirectly regulates the stability of BRCA1 via other interacting proteins.”

Q2. Another issue arises from the interpretation of the newly added data and the lack of controls for isolated UNE-I interacting with and stabilizing BRCA1 (Fig. 4b, e): In their revised discussion (lines 443-447) and point-by-point response (p. 2), the authors claim that the stabilizing effect of overexpressed UNE-I on BRCA1 must be direct, which should rule out an indirect effect by destabilizing or altering the MSC composition. However, as the authors show in Figure 1a, overexpressed UNE-I efficiently interacts with EPRS in the cell, suggesting that it competes with full-length IleRS for incorporation into the MSC. Thus, full-length IleRS would be released from the MSC, possibly together with other components (as the authors point out: “...full-length IleRS is required for the stability of MSC as the N-terminal editing, catalytic and UNE-I domains contribute to the assembly of MSC”. **Since the knock-down of endogenous IleRS seems to be inefficient in Figures 4b and e, the observed effects on BRCA1 may still be indirect.** Thus, the authors should include controls to show that the positive effect of isolated UNE-I on BRCA1 stability is not due to increased levels of free and nuclear-localized full-length IleRS or due to a general destabilization of the MSC by incorporation of overexpressed UNE-I instead of full-length IleRS.

>> Since we used the NLS-sequence fused to UNE-I in Fig 4b experiment (line 325, 327 P11), we expect a significant fraction of the NLS-UNE-I to be located to the nucleus. Even if a small fraction of NLS-UNE-I can incorporate to MSC, the amount of IleRS released from MSC would not be high enough to affect the stability of nuclear BRCA1. In Fig 3e (left panel), we showed the expression levels of NLS-IleRS, -UNE-I or -IleRS (Δ UNE-I) in whole cell lysates (WCL). The BRCA1 level appears to be higher in the UNE-I-expressed cells compared to those in the IleRS (Δ UNE-I)- or WT IleRS-expressed cells and this result is consistent with the fact that the amount of BRCA1 immunoprecipitated is higher with NLS-UNE-I than with NLS-WT or -IleRS (Δ UNE-I) (right panel).

Fig 3e

Fig 4b

To further test our idea, we overexpressed UNE-I (and NLS-UNE-I) and examined the level of IleRS in the nucleus. The level of nuclear IleRS in the cells with overexpressed UNE-I was similar to the level of normal cells. Thus, overexpressed UNE-I did not affect localization of IleRS to the nucleus.

Q3: Figure 3h: The authors should include negative controls in their BRCA1 pull-down, which could ideally include AIMP3 or EPRS, both of which the authors show to be not involved in nuclear stabilization of BRCA1.

>> We appreciate the reviewer's comments. As we mentioned earlier, AIMP3 or EPRS can also bind to BRCA1 as IleRS (Ertych et al. 2016 PNAS <https://doi.org/10.1073/pnas.1525129113>). However, binding itself does not warrant the same effect and may result in different functional outcome. For instance, one of the MSC components, AIMP2, can bind and stabilize p53 in nucleus by blocking MDM2-mediated ubiquitination (Han, PNAS 105, 11206, 2008). However, its splicing variant, AIMP2-DX2, can also bind to p53 and compete with its native form, AIMP2. Yet, binding of AIMP2-DX2 stimulate MDM2-mediated ubiquitination of p53 (Choi, Plos Genetics 7, e1001351, 2011). We thus thought that IgG is more desirable as negative control to avoid complicated interpretation by using other potential BRCA1 binders.

Q4: Line 116: The authors included “less efficiently” in the sentence “By contrast, the UNE-I domain alone associated most prominently with GluProRS and less efficiently with GlnRS, ArgRS, and AspRS, but did not interact with LeuRS.” It is not clear how they reach this conclusion from their mass-spec results, as there are more peptides of AspRS and even LysRS pulled down by UNE-I than EPRS (relative to IleRS). It would be important to know what peptides are listed in Fig. 1a (i.e. absolute numbers or unique peptides), as their number clearly depends on the size of the proteins.

>> We thank a reviewer for this comment. Please see the revised Fig 1a. We have revised the figure to provide the information more clearly. We have deposited the detailed mass analysis data (seq and numbers) in the database (A reviewer token/link (PXD031261 and 10.6019/PXD031261) <https://www.ebi.ac.uk/pride/login>;

Username: reviewer_pxd031261@ebi.ac.uk; Password: RSpWE3t2). We also changed the sentence (line 116, P5) to “...By contrast, the UNE-I domain alone associated with GluProRS, GlnRS, ArgRS, and AspRS, but did not interact with LeuRS.”

Response to reviewer #3's comments

The authors have satisfactorily addressed all my concerns. The revised manuscript is a great improvement on the original submission and was a pleasure to read. I have only a few minor points that the authors might wish to take note of:

1. Pg. 8, line 212 - "UXB2" should be changed to UBX2.

We have corrected the typo.

2. Pg. 9, line 233 - refer to Figure 2c only instead of Figure 2b and 2c.

We have removed “Figure 2b” in the citation.

3. In the legend of Figure 2 - mention that "WCL" is whole cell lysate as some readers may not be familiar with this acronym.

>> We revised Fig 2f with WCL (Whole Cell Lysate). We thank a reviewer for this comment.

REVIEWERS' COMMENTS

Reviewer #1 (Remarks to the Author):

The authors have addressed all my comments and concerns in their revised manuscript. I therefore recommend its publication.